# MCM2–7-dependent cohesin loading during S phase promotes sister-chromatid cohesion

Ge Zheng[1], Mohammed Kanchwala[2], Chao Xing[2,3,4], Hongtao Yu[1]*

[1]Howard Hughes Medical Institute, Department of Pharmacology, University of Texas Southwestern Medical Center, Dallas, United States; [2]Bioinformatics Lab, Eugene McDermott Center for Human Growth and Development, University of Texas Southwestern Medical Center, Dallas, United States; [3]Department of Clinical Sciences, University of Texas Southwestern Medical Center, Dallas, United States; [4]Department of Bioinformatics, University of Texas Southwestern Medical Center, Dallas, United States

**Abstract** DNA replication transforms cohesin rings dynamically associated with chromatin into the cohesive form to establish sister-chromatid cohesion. Here, we show that, in human cells, cohesin loading onto chromosomes during early S phase requires the replicative helicase MCM2–7 and the kinase DDK. Cohesin and its loader SCC2/4 (NIPBL/MAU2 in humans) associate with DDK and phosphorylated MCM2–7. This binding does not require MCM2–7 activation by CDC45 and GINS, but its persistence on activated MCM2–7 requires fork-stabilizing replisome components. Inactivation of these replisome components impairs cohesin loading and causes interphase cohesion defects. Interfering with Okazaki fragment processing or nucleosome assembly does not impact cohesion. Therefore, MCM2–7-coupled cohesin loading promotes cohesion establishment, which occurs without Okazaki fragment maturation. We propose that the cohesin–loader complex bound to MCM2–7 is mobilized upon helicase activation, transiently held by the replisome, and deposited behind the replication fork to encircle sister chromatids and establish cohesion.
DOI: https://doi.org/10.7554/eLife.33920.001

*For correspondence:
hongtao.yu@utsouthwestern.edu

**Competing interests:** The authors declare that no competing interests exist.

## Introduction

Sister-chromatid cohesion is essential for proper chromosome segregation and faithful transmission of the genome during the cell cycle (*Morales and Losada, 2018*; *Uhlmann, 2016*). Failure to establish or resolve cohesion in a timely manner leads to genomic instability and aneuploidy. Sister-chromatid cohesion is mediated by cohesin, a ring-shaped ATPase machine that consists of SMC1A, SMC3, RAD21, and either STAG1 or STAG2 in human somatic cells (*Haarhuis et al., 2014*; *Losada and Hirano, 2005*; *Nasmyth and Haering, 2009*; *Onn et al., 2008*; *Peters et al., 2008*; *Zheng and Yu, 2015*). Cohesin rings topologically entrap DNA to generate physical linkages between sister chromatids and enable cohesion. Cohesin regulates other chromosome-based processes, such as DNA repair, transcription, and chromosome folding (*Merkenschlager and Odom, 2013*; *Wu and Yu, 2012*). These other functions of cohesin likely also involve the topological entrapment of chromosomes or possibly the extrusion of DNA loops (*Barrington et al., 2017*; *Davidson et al., 2016*; *Haarhuis et al., 2017*).

Cohesin is loaded onto chromosomes in telophase and G1 by the SCC2/4 complex (NIPBL/MAU2 in humans) (*Ciosk et al., 2000*; *Gillespie and Hirano, 2004*; *Takahashi et al., 2004*; *Tonkin et al., 2004*; *Watrin et al., 2006*). Before DNA replication, the chromosome-bound cohesin is dynamic and is actively removed from chromosomes by the cohesin-releasing factor WAPL with the help of the

scaffolding protein PDS5A or PDS5B (*Chan et al., 2012*; *Kueng et al., 2006*; *Lopez-Serra et al., 2013*; *Ouyang and Yu, 2017*; *Ouyang et al., 2013*; *Ouyang et al., 2016*). During DNA replication in S phase, a pool of cohesin is converted to the cohesive form, which stably associates with chromosomes and mediates sister-chromatid cohesion (*Gerlich et al., 2006*; *Kueng et al., 2006*). In human cells, cohesion establishment requires the acetylation of SMC3 by the acetyltransferases ESCO1 and ESCO2 and subsequent recruitment of sororin, which antagonizes WAPL to stabilize cohesin on chromosomes (*Alomer et al., 2017*; *Hou and Zou, 2005*; *Nishiyama et al., 2010*; *Ouyang et al., 2016*; *Rankin et al., 2005*; *Rolef Ben-Shahar et al., 2008*; *Rowland et al., 2009*; *Unal et al., 2008*; *Zhang et al., 2008a*).

The replicative helicase MCM2–7 is loaded on chromosomes in telophase with the help of the origin recognition complex (ORC), CDC6, and CDT1, and exists as dormant double hexamers (*Bell and Labib, 2016*; *Riera et al., 2017*). At the G1/S transition, phosphorylation of MCM2–7 by the DBF4-dependent kinase (DDK) and CDKs promotes the formation of the active CDC45–MCM–GINS (CMG) helicase capable of unwinding DNA. DNA polymerases and additional replication factors are further recruited to replication forks to assemble the replisome and initiate DNA replication. Downregulation of a subset of replisome components, such as Ctf4, Tof1, Csm3, or Chl1 in yeast and their corresponding orthologs WDHD1, TIMELESS, TIPIN, or DDX11 in metazoans, permits DNA replication, but produces cohesion defects (*Chan et al., 2003*; *Errico et al., 2009*; *Leman et al., 2010*; *Lengronne et al., 2006*; *Parish et al., 2006*; *Rudra and Skibbens, 2013*; *Samora et al., 2016*; *Sherwood et al., 2010*; *Tanaka et al., 2009*). Thus, cohesion establishment is tightly coupled to DNA replication, but the mechanism of this coupling is incompletely understood.

Initial experiments in yeast showed that inactivation of the Scc2/4 complex in cells enriched in early S phase by hydroxyurea did not reduce cell viability or cause overt cohesion defect, suggesting that cohesion establishment might not require de novo cohesin loading in S phase (*Lengronne et al., 2006*). This result led to the hypothesis that cohesion establishment involves the entire replication machinery passing through the cohesin rings pre-loaded on DNA in G1. Consistent with this hypothesis, in WAPL-deficient human cells, cohesin loaded in G1 remains associated with chromatin throughout S phase and is not displaced by the replication machinery (*Rhodes et al., 2017b*). Recent evidence in yeast, however, indicates that the Scc2/4 complex is still needed in S phase for cohesion establishment (*Murayama et al., 2018*; *Nasmyth, 2017*). In vitro reconstitution experiments further suggest that, with the help of Scc2/4, cohesin loaded on DNA can further capture a second DNA molecule, which has to be single-stranded (*Murayama et al., 2018*). These results suggest that de novo loading of cohesin at the replication fork, where single-stranded DNA is present, might promote cohesion establishment.

In *Xenopus* egg extracts, Scc2/4-dependent cohesin loading onto chromosomes requires MCM2–7 and DDK, which physically interact with Scc2/4 (*Gillespie and Hirano, 2004*; *Takahashi et al., 2008*; *Takahashi et al., 2004*). Activation of the MCM2–7 helicase and initiation of DNA replication are not required for cohesin loading or Scc2/4 interaction with MCM and DDK, indicating that Scc2/4 and cohesin interact with the pre-replication complex (pre-RC). Although the functional consequence of this interaction in cohesion establishment was not directly examined, these findings suggested an attractive mechanism that couples cohesin loading to the DNA replication machinery. On the other hand, a subsequent study showed that MCM2–7 might be dispensable for cohesin loading in human cells, although the MCM–cohesin interaction could be detected (*Guillou et al., 2010*). Moreover, Cdc6 and, by inference, the pre-RC are not required for cohesin loading in yeast (*Uhlmann and Nasmyth, 1998*). These results casted doubts on the conservation of MCM-dependent cohesin loading in organisms other than *Xenopus*.

To gain insight into cohesion establishment during DNA replication, we investigated the requirements of various replication factors in cohesin loading and interphase cohesion in human cells. Our results show that MCM2–7 and DDK physically interact with NIPBL/MAU2 and cohesin, and are required for cohesin loading in human cells during early S phase. Depletion of replisome components causes cohesion defects in interphase and reduces the MCM–NIPBL–cohesin interaction. Thus, binding of NIPBL/MAU2 and cohesin to MCM persists at active replication forks, requires additional replisome components for maintenance, and is critical for cohesion establishment. We propose that cohesin and NIPBL/MAU2 loaded at replication origins remain associated with MCM2–7 and DDK, and are mobilized upon replication initiation, held transiently by replisome components, and deposited behind the replication fork to establish sister-chromatid cohesion.

## Results

### Cohesin loading requires MCM2–7 in human cells during early S phase

Using RNA interference (RNAi), we depleted MCM2–7 components from HeLa Tet-On cells stably expressing RAD21-Myc and examined the level of chromatin-bound RAD21-Myc with immunostaining. RAD21-Myc is functional, as it could rescue cohesion defects caused by depletion of the endogenous RAD21 (*Wu et al., 2012*). In cells arrested in early S phase with thymidine treatment, the intensity of RAD21-Myc on chromatin was greatly reduced when MCM2 or NIPBL was depleted (*Figure 1A,B*). The depletion of MCM2 was very efficient (*Figure 1C*). The RAD21-Myc signal on chromatin was restored when RNAi-resistant MCM2 was ectopically expressed (*Figure 1—figure supplement 1A*). In fact, MCM2 overexpression elevated the RAD21-Myc intensity on chromatin to a level higher than that in control RNAi cells. We thus checked whether other MCM subunits were required for cohesin loading. The RAD21-Myc intensity in early S phase cells was similarly reduced by the depletion of MCM3 or MCM5 (*Figure 1—figure supplement 1B*). We next examined the level of MCM5 on chromatin when MCM2 was overexpressed. MCM2 overexpression elevated MCM5 levels on chromatin (*Figure 1—figure supplement 1C,D*). Thus, MCM2 overexpression likely increases the amount of the MCM2–7 complex on chromatin, resulting in hyperactive cohesin loading. These results suggest that the MCM2–7 complex is required for cohesin loading in human cells during early S phase.

In contrast to thymidine-arrested S phase cells, depletion of MCM2 only marginally reduced the RAD21-Myc intensity on chromatin in telophase cells, indicating that MCM2–7 is not strictly required for cohesin loading in telophase (*Figure 1D,E*). Depletion of NIPBL greatly reduced chromatin-bound RAD21-Myc in both cell cycle stages. MCM2 depletion also reduced the association of endogenous STAG2 with chromatin in thymidine-treated cells (*Figure 1—figure supplement 1E*), but only slightly reduced chromatin-bound STAG2 in telophase cells (*Figure 1—figure supplement 1F*). Thus, MCM2–7 is more critical for NIPBL/MAU2-dependent cohesin loading during early S phase.

We next examined the interactions among MCM2–7, cohesin, and NIPBL/MAU2 at different cell cycle stages. MCM2 associated with both NIPBL/MAU2 and cohesin in log-phase and thymidine-arrested, early S phase cells, but this interaction was much weaker in G2 or mitotic cells (*Figure 1F*). To narrow down the time window of this MCM–NIPBL–cohesin interaction, we collected cells that were released from thymidine arrest at different time points and performed MCM2 immunoprecipitation. The MCM–NIPBL–cohesin interaction reached peak levels when cells were enriched in mid-S phase at 2 hr after thymidine release (*Figure 1G,H*). Importantly, all lysates used for immunoprecipitation were treated with Turbo nuclease. The observed MCM–NIPBL–cohesin interaction was unlikely to be bridged by chromatin. The cohesion-stabilizing factor sororin was also detected in the MCM2 IP, indicating that at least a pool of MCM-bound cohesin was capable of establishing cohesion.

We tested whether NIPBL/MAU2 binding to MCM2–7 was dependent on cohesin and vice versa. Depletion of NIPBL/MAU2 inhibited the association of cohesin with MCM2 (*Figure 2A*). When SMC3 was depleted from cells, the interaction between MCM2 and NIPBL/MAU2 was no longer detectable. Thus, NIPBL/MAU2 binding and cohesin binding to MCM2–7 are mutually dependent. NIPBL/MAU2 and cohesin may bind cooperatively to MCM2–7. Alternatively, although this interaction can exist without chromatin, it may only be established on chromatin, following NIPBL/MAU2-dependent cohesin loading. The protein levels of NIPBL/MAU2 were reduced in SMC3 RNAi cells, suggesting that most NIPBL/MAU2 molecules in the cell were bound to cohesin.

We next determined the genome-wide distribution of MCM2, NIPBL/MAU2, and cohesin in thymidine-arrested cells, using chromatin immunoprecipitation followed by next generation sequencing (ChIP-seq) (*Figure 2B,C*). There were over 50,000 RAD21-enriched peaks in the genome. As expected, the top RAD21-binding consensus motif matched that of the chromatin insulator CTCF (*Figure 2—figure supplement 1A*). Comparison with the CTCF ChIP-seq data in the ENCODE database revealed that RAD21-enriched peaks were frequently co-occupied by CTCF (*Figure 2—figure supplement 1B*). In contrast, there were very few NIPBL peaks in our ChIP-seq dataset (*Figure 2B, C*). Consistent with previous reports (*Kagey et al., 2010*), more than 50% of these NIPBL peaks were located in promoter regions (*Figure 2—figure supplement 2*). The majority of these NIPBL peaks were co-occupied by cohesin in two separate experiments (*Figure 2B,C*), consistent with the notion that NIPBL remained bound to cohesin on chromatin (*Rhodes et al., 2017a*). We only

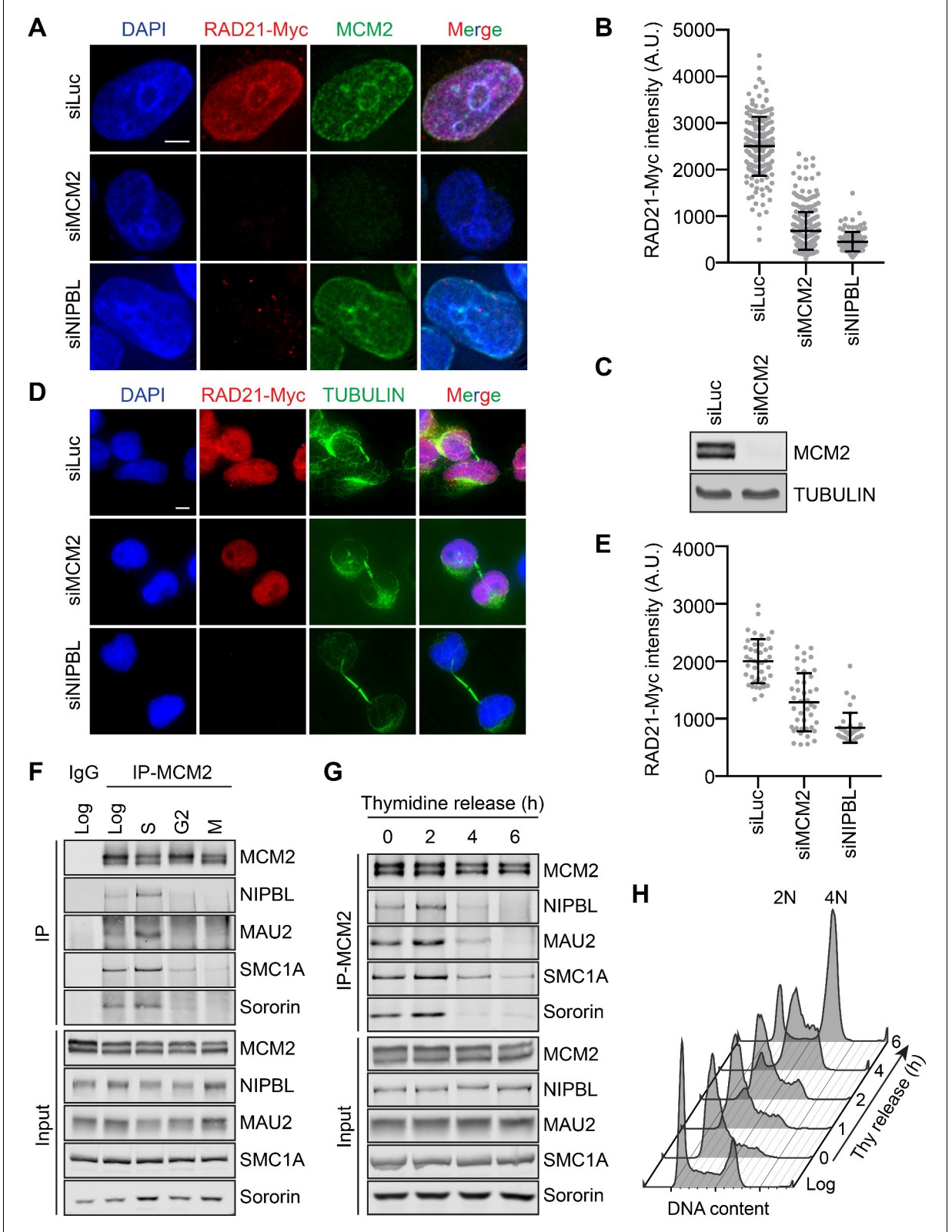

**Figure 1.** The MCM2–7 complex is required for cohesin loading during early S phase. (**A**) DAPI (blue), anti-Myc (red), and anti-MCM2 (green) staining of HeLa cells that stably expressed RAD21-Myc. Cells were transfected with the indicated siRNAs and arrested in early S phase with thymidine before fixation and staining. Scale bar, 5 μm. (**B**) Quantification of the chromatin intensities of RAD21-Myc of cells in (**A**). Each dot in the graph represents a single cell. Mean ± SD (siLuc, n = 184; siMCM2, n = 295; siNIPBL, n = 115). (**C**) Lysates of HeLa cells either mock transfected or transfected with siMCM2

*Figure 1 continued on next page*

*Figure 1 continued*

were blotted with the indicated antibodies. (**D**) DAPI (blue), anti-Myc (red), and anti-TUBULIN (green) staining of telophase HeLa cells that stably expressed RAD21-Myc. Cells were transfected with the indicated siRNAs and released from nocodazole-mediated mitotic arrest for 4 hr before fixation. Scale bar, 5 µm. (**E**) Quantification of the RAD21-Myc chromatin intensities of cells in (**D**). Each dot in the graph represents a single cell. Mean ± SD (siLuc, n = 42; siMCM2, n = 42; siNIPBL, n = 38). (**F**) Log-phase HeLa cells and cells synchronized in early S phase by thymidine, G2 by the CDK1 inhibitor, or mitosis by nocodazole were collected and lysed in the presence of nuclease. The total cell lysates (input) and anti-MCM2 immunoprecipitate (IP) were blotted with the indicated antibodies. IgG IP from log-phase cells was used as a negative control. (**G**) HeLa cells were synchronized with thymidine for 16–18 hr, released from thymidine, and harvested at the indicated time points. Cells were lysed in the presence of nuclease. The total cell lysates (input) and anti-MCM2 IP were blotted with the indicated antibodies. (**H**) Flow cytometry analysis of log-phase HeLa cells and cells released from the thymidine arrest for the indicated times.

DOI: https://doi.org/10.7554/eLife.33920.002

The following figure supplement is available for figure 1:

**Figure supplement 1.** MCM2–7 promotes cohesin loading in early S phase human cells.

DOI: https://doi.org/10.7554/eLife.33920.003

performed MCM2 ChIP-seq once (Experiment 1). The MCM2 ChIP-seq signals were broadly distributed throughout the genome, with no discernable peaks of enrichment. Therefore, the MCM2–NIPBL–cohesin complex likely also lacks defined distributions in the genome, providing a possible explanation for the paucity of NIPBL peaks. We next used a clustering approach called SICER to identify MCM2-enriched domains (***Figure 2—figure supplement 1B***). When NIPBL peaks were used as the reference, these MCM2-enriched domains and RAD21 peaks were found to co-localize with NIPBL peaks (***Figure 2—figure supplement 1C***). Genome-wide co-localization of MCM2–7 and cohesin in human cells has also recently been reported (***Cucco et al., 2018***). Because cohesin is more abundant than NIPBL/MAU2, cohesin not associated with NIPBL/MAU2 and MCM2-7 is likely enriched at CTCF sites.

Taken together, our results indicate that the MCM2–7 complex is required for cohesin loading during S phase, and MCM2–7, NIPBL/MAU2, and cohesin physically interact in a DNA-independent manner during DNA replication.

## DDK is required for cohesin loading and the MCM–NIPBL–cohesin interaction

Consistent with the findings in *Xenopus* egg extracts (***Takahashi et al., 2008***), cohesin association with chromosomes was greatly reduced when CDC7, the catalytic subunit of DDK, was depleted in human cells arrested in early S phase by thymidine (***Figure 3A*** and ***Figure 3—figure supplement 1A,B***). Chromatin-bound cohesin was less affected by CDC7 depletion in telophase cells (***Figure 3—figure supplement 1C,D***). Depletion of CDC7 was efficient, and greatly reduced the phosphorylated, fast-migrating form of MCM2 (***Figure 3B***), as did lambda phosphatase (λPPase) treatment (***Figure 3C***). We noticed that the effects of MCM2, NIPBL, or CDC7 depletion on the chromatin association of RAD21-Myc were greater than those on the chromatin binding of the endogenous STAG2, particularly in S phase cells. The underlying reason for this observation is unclear, but might be due to NIPBL/MAU2-independent chromatin association of STAG2 or due to trivial technical issues. For example, RAD21-Myc might be partially defective for MCM-independent loading mechanisms.

Consistent with its effect on cohesin loading, CDC7 depletion in early S phase cells reduced the binding of NIPBL/MAU2, cohesin, and sororin to MCM2 (***Figure 3B***). DDK is a heterodimer consisting of CDC7 and a regulatory subunit, DBF4 or DRF1, in human cells. Depletion of either DBF4 or DRF1 partially reduced the phosphorylation of MCM2 and the MCM–NIPBL–cohesin interaction (***Figure 3—figure supplement 2A***). Depletion of both reduced to a greater extent the phosphorylation of MCM2, and MCM2 binding to CDC7, NIPBL/MAU2, and cohesin. Therefore, the intact DDK is required for cohesin association with chromatin and for the MCM–NIPBL–cohesin interaction in human cells during early S phase.

NIPBL preferentially bound to the fast-migrating, phosphorylated form of MCM2 (***Figure 3—figure supplement 2B***), suggesting that phosphorylation of MCM2–7 by DDK might be required for the MCM–NIPBL–cohesin interaction. Indeed, XL413, a specific chemical inhibitor of the CDC7 kinase activity, effectively blocked the phosphorylation of MCM2 in early S phase cells (***Figure 3D***),

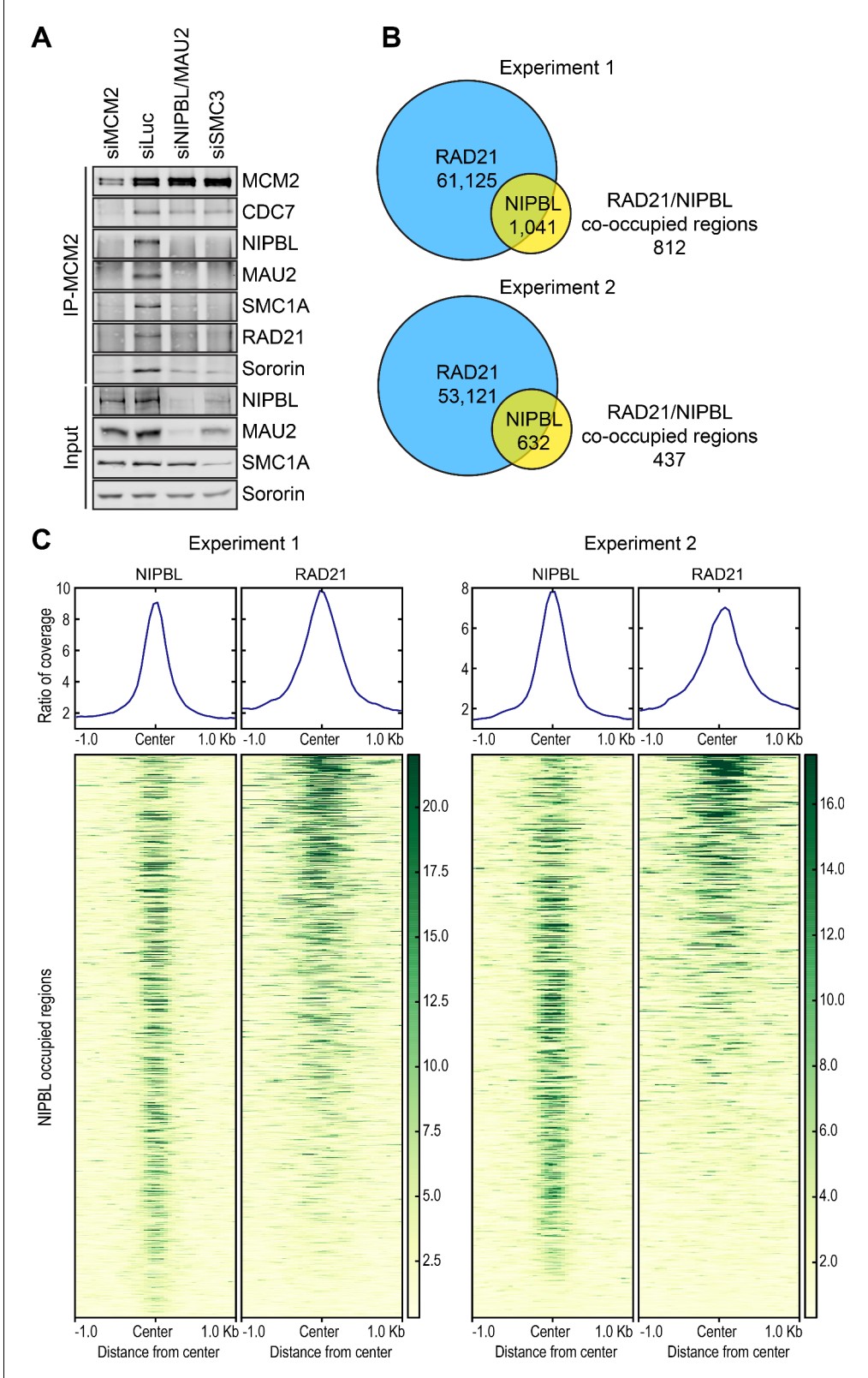

**Figure 2.** NIPBL and cohesin bind to MCM2–7 interdependently and co-localize in the genome. (**A**) HeLa cells were transfected with the indicated siRNAs, arrested in early S phase with thymidine, and lysed in the presence of nuclease. The total lysates (input) and anti-MCM2 IP were blotted with the indicated antibodies. (**B**) Venn diagrams showing the overlap of cohesin (RAD21)-occupied sites with those bound by NIPBL in two independent

*Figure 2 continued on next page*

*Figure 2 continued*

ChIP-seq experiments. (**C**) Region map showing that, at the majority of the NIPBL-occupied regions, the occupancy of RAD21 was detected reproducibly in the two ChIP-seq experiments.

DOI: https://doi.org/10.7554/eLife.33920.004

The following figure supplements are available for figure 2:

**Figure supplement 1.** ChIP-seq analysis of cohesin, NIPBL, and MCM2 in human cells.

DOI: https://doi.org/10.7554/eLife.33920.005

**Figure supplement 2.** ChIP-seq analysis of cohesin, NIPBL, and MCM2 in human cells.

DOI: https://doi.org/10.7554/eLife.33920.006

and greatly reduced the level of chromatin-bound cohesin, without affecting MCM2 loading (*Figure 3E,F and Figure 3—figure supplement 3A–D*). The MCM–NIPBL–cohesin interaction in cells treated with XL413 was reduced to an extent similar to that in cells depleted of CDC7 (*Figure 3G*), while the interaction between MCM2 and CDC7 remained intact in XL413-treated cells. In CDC7-depleted or XL413-treated cells, the reduction of cohesin binding to MCM2 was not as dramatic as the reduction of NIPBL/MAU2 binding to MCM2. Consistent with the phenotypes of CDC7 depletion, XL413 did not significantly inhibit cohesin loading in telophase cells (*Figure 3—figure supplement 3E,F*). Our results are in general agreement with the published findings in *Xenopus* (*Takahashi et al., 2008*), and indicate that DDK-mediated phosphorylation of MCM2–7 and possibly other proteins is required for cohesin loading and for NIPBL/MAU2 binding to MCM2–7 during early S phase.

We tested whether activation of the MCM2–7 helicase and replication initiation were required for MCM-dependent cohesin loading and the integrity of the MCM–NIPBL–cohesin interaction. Chromatin-bound cohesin and the binding of NIPBL/MAU2 and cohesin to MCM2 were not decreased when CDC45 or GINS1 was depleted (*Figure 3—figure supplement 4A–C*). Depletion of CDC45 or GINS1 was efficient (*Figure 3—figure supplement 4D*), and slowed S phase progression as evidenced by an increased percentage of cells incorporating BrdU but a reduced extent of BrdU incorporation per cell (*Figure 3—figure supplement 4E*). These results suggest that CDC45 and GINS are dispensable for cohesin loading and stable MCM–NIPBL–cohesin interaction, again consistent with findings in *Xenopus* (*Gillespie and Hirano, 2004*; *Takahashi et al., 2004*). Because the MCM–NIPBL–cohesin interaction peaks in mid-S phase cells (*Figure 1G*), helicase activation and DNA replication likely do not block this interaction. Thus, the formation of the MCM–NIPBL–cohesin complex is independent of helicase activation and replication initiation, and this complex likely persists on active replication forks.

## Replisome components promote the establishment of sister-chromatid cohesion

To examine cohesion status in interphase cells immediately after DNA replication, we developed a fluorescence in situ hybridization (FISH) assay, using a probe (466L19) that specifically recognized a locus on chromosome 3. After depletion of replisome components individually or in combination, we collected cells that were released from thymidine for 4 hr and measured the distance between paired FISH dots in late S/G2 phase cells. Expectedly, sororin depletion caused strong cohesion defects (*Figure 4A*), as evidenced by the greatly increased distances between the paired FISH signals.

Several replisome components were also required for cohesion establishment (*Figure 4A,B*). In particular, MCM2 depletion caused obvious cohesion defect. Depletion of WDHD1, TIMELESS, TIPIN, or DDX11 produced mild cohesion defect. Strikingly, co-depletion of WDHD1 and TIMELESS caused strong cohesion defects, whereas co-depletion of WDHD1/DDX11 or TIMELESS/DDX11 did not produce synergistic effects. Consistent with the FISH assay, the interaction between cohesin and sororin was greatly reduced in cells co-depleted of WDHD1 and TIMELESS, but was only modestly reduced with each single depletion (*Figure 4—figure supplement 1A*). Expression of siRNA-resistant Myc-WDHD1 or Myc-TIMELESS largely rescued the cohesion defect in cells co-depleted of WDHD1 and TIMELESS (*Figure 4—figure supplement 1B,C*), ruling out siRNA off-target effects. These results confirm that replisome components promote cohesion establishment in human cells

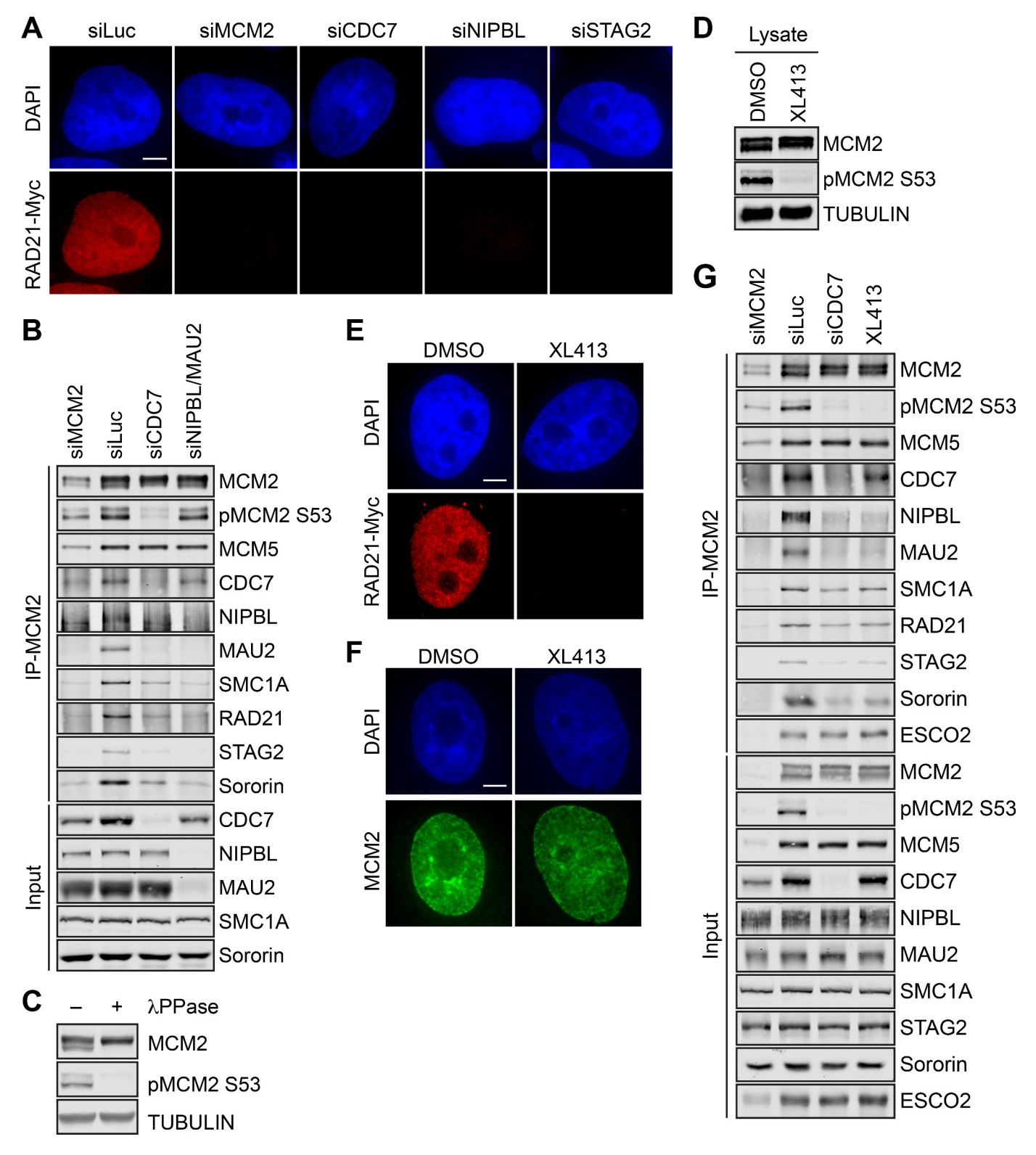

**Figure 3.** DDK promotes the MCM–NIPBL–cohesin interaction. (**A**) DAPI (blue) and anti-Myc (red) staining of HeLa cells that stably expressed RAD21-Myc. Cells were transfected with the indicated siRNAs and arrested in early S phase with thymidine. Scale bar, 5 µm. (**B**) Lysates of HeLa cells transfected with the indicated siRNAs and synchronized in early S phase were treated with Turbo nuclease and immunoprecipitated with anti-MCM2. The total lysates (input) and anti-MCM2 immunoprecipitate (IP) were blotted with the indicated antibodies. (**C**) Lysates of HeLa cells were incubated

*Figure 3 continued on next page*

*Figure 3 continued*

with or without λPPase and blotted with the indicated antibodies. (**D**) Lysates of HeLa cells treated DMSO or the DDK kinase inhibitor XL413 (dissolved in DMSO) were blotted with the indicated antibodies. (**E,F**) DAPI (blue), anti-Myc (red), and anti-MCM2 (green) staining of RAD21-Myc-expressing HeLa cells that were treated with DMSO or XL413 and arrested in early S phase by thymidine. Scale bar, 5 μm. (**G**) HeLa cells were either transfected with the indicated siRNAs or treated with XL413, arrested in early S phase by thymidine, and lysed in the presence of Turbo nuclease. The total lysates (input) and anti-MCM2 IP were blotted with the indicated antibodies.

DOI: https://doi.org/10.7554/eLife.33920.007

The following figure supplements are available for figure 3:

**Figure supplement 1.** DDK promotes cohesin loading in early S phase, but not in telophase.

DOI: https://doi.org/10.7554/eLife.33920.008

**Figure supplement 2.** DDK promotes the MCM–NIPBL–cohesin interaction in early S phase.

DOI: https://doi.org/10.7554/eLife.33920.009

**Figure supplement 3.** The kinase activity of DDK is critical for cohesin loading in early S phase, but not in telophase.

DOI: https://doi.org/10.7554/eLife.33920.010

**Figure supplement 4.** CDC45 and GINS are dispensable for cohesin loading and the MCM–NIPBL–cohesin interaction.

DOI: https://doi.org/10.7554/eLife.33920.011

and they may do so in different pathways, as suggested by findings in yeast (*Borges et al., 2013*). On the other hand, as each RNAi-mediated depletion is incomplete, depletion of different components in the same pathway might also result in a more complete inactivation of that pathway.

WAPL depletion rescued the cohesion defects caused by WDHD1 and TIMELESS co-depletion (*Figure 4—figure supplement 1B,D*), suggesting that replisome components might contribute to cohesion establishment through antagonizing WAPL in human cells. The WAPL effect is different from previous findings in yeast, where Wapl inactivation does not bypass the cohesion defects caused by mutations in certain replisome components (*Borges et al., 2013*). Again, RNAi-mediated depletion is incomplete. It remains to be determined whether cohesion defects caused by the complete loss of WDHD1 and TIMELESS can be bypassed by WAPL inactivation in human cells.

We did not observe strong cohesion defects with NIPBL depletion, even though chromatin-bound cohesin was greatly reduced when NIPBL was silenced during early S phase. Similarly, depletion of CDC7, CDC45, or GINS1 did not produce strong cohesion defects. We noticed that a large percentage of cells depleted of NIPBL, CDC7, CDC45, or GINS1 had two single FISH dots each (as opposed to two pairs of juxtaposed dots) (*Figure 4C*), suggesting that this locus was not replicated in these cells at 4 hr after the release from thymidine. Moreover, control cells replicated their DNA and slowly progressed through S phase during prolonged thymidine treatment for 48 hr (*Figure 4—figure supplement 1E*). This highly inefficient S phase progression was inhibited when NIPBL was depleted. In cases of DDK, CDC45, or GINS1 depletion, the impaired replication was expected, because of the known roles of these proteins in establishing the active CMG helicase. The requirement for NIPBL in DNA replication was not entirely expected. We do not know whether this requirement was due to a perturbation of the transcription function of cohesin in G1 or direct roles of NIPBL/MAU2 and cohesin in DNA replication and repair. Regardless, our results suggest that the strict requirement for these factors in DNA replication might explain the lack of strong cohesion defects in cells depleted of these factors. Cells with a more complete depletion of these factors cannot initiate DNA replication. Only those with a partial depletion can replicate DNA to some extent and manifest dampened cohesion defects.

## Cohesion establishment is independent of lagging strand maturation and histone deposition

We next tested whether all replication factors were involved in cohesion establishment. Flap endonuclease 1 (FEN1) processes the 5' ends of Okazaki fragments during lagging strand DNA synthesis. DNA ligase 1 (LIG1) then joins the processed Okazaki fragments. CHAF1A mediates histone deposition and chromatin assembly during DNA replication. We did not observe overt cohesion defects in S/G2 phase cells depleted of FEN1, LIG1, or CHAF1A (*Figure 4D,E*). Depletion of LIG1 or CHAF1A did not increase the percentage of cells with an unreplicated chromosome three locus (*Figure 4C*). Thus, not all replication factors are required for interphase cohesion in human cells. Our results further suggest that cohesion establishment occurs prior to or independently of the processing and

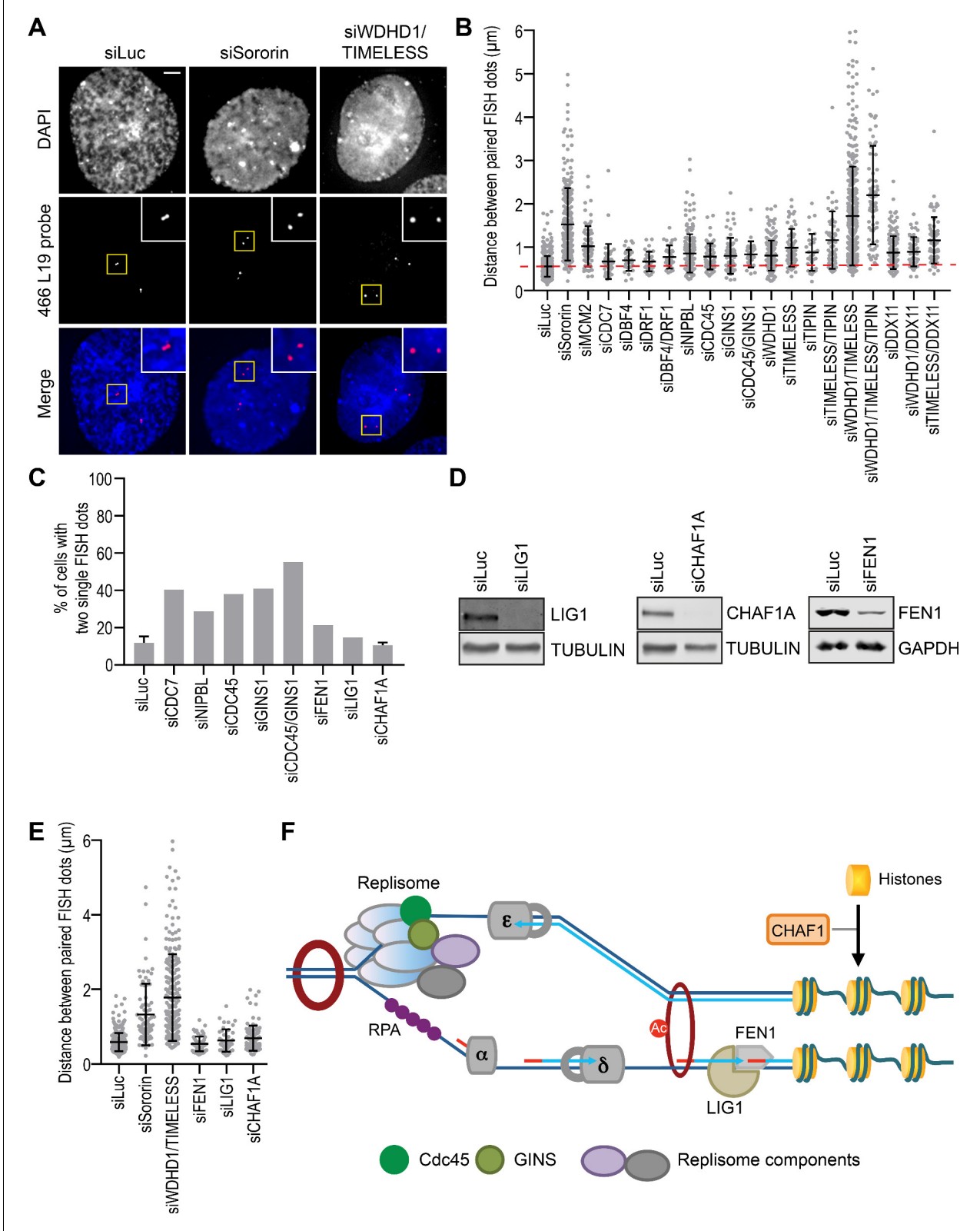

**Figure 4.** Replisome components are required for interphase sister-chromatid cohesion. (**A**) Representative images of G2-enriched HeLa cells transfected with the indicated siRNAs and stained with DAPI (blue in merge) and the FISH probe (red in merge). Cells were treated with thymidine for 16–18 hr and released into fresh medium for 4 hr before fixation. Selected paired FISH signals are magnified in inset. Scale bar, 5 µm. (**B**) Quantification of the distances between paired FISH signals in G2-enriched HeLa cells transfected with the indicated siRNAs. Mean ± SD (siLuc, n = 502; siSororin,

*Figure 4 continued on next page*

*Figure 4 continued*

n = 243; siMCM2, n = 70; siCDC7, n = 44; siDBF4, n = 30; siDRF1, n = 30; siDBF4/DRF1, n = 32; siNIPBL, n = 191; siCDC45, n = 66; siGINS1, n = 68; siCDC45/GINS1, n = 26; siWDHD1, n = 198; siTIMELESS, n = 58; siTIPIN, n = 31; siTIMELESS/TIPIN, n = 51; siWDHD1/TIMELESS, n = 375; siWDHD1/TIMELESS/TIPIN, n = 71; siDDX11, n = 154; siWDHD1/DDX11, n = 52; siTIMELESS/DDX11, n = 65). The red dashed line indicates the mean of the siLuc sample. (C) Quantification of the percentage of cells with two unreplicated single FISH dots. Cells were transfected with the indicated siRNAs, treated with thymidine for 16–18 hr, and released into fresh medium for 4 hr. siLuc, n = 507 (mean ± SD; four independent experiments); siCDC7, n = 109; siNIPBL, n = 66; siCDC45, n = 71; siGINS1, n = 139; siCDC45/GINS1, n = 116; siFEN1, n = 112; siLIG1, n = 115; siCHAF1A, n = 226 (mean ± SD; two independent experiments). (D) Lysates of HeLa cells transfected with the indicated siRNAs were blotted with the indicated antibodies. (E) Quantification of the distances between paired FISH signals in G2-enriched HeLa cells transfected with the indicated siRNAs. Mean ± SD (siLuc, n = 382; siSororin, n = 82; siWDHD1/TIMELESS, n = 226; siFEN1, n = 60; siLIG1, n = 44; siCHAF1A, n = 110). (F) A simplified model delineating the molecular events during DNA replication. Cohesion establishment can occur without the maturation of Okazaki fragments and the deposition of histones.
DOI: https://doi.org/10.7554/eLife.33920.012

The following figure supplement is available for figure 4:

**Figure supplement 1.** Replisome components promote sister-chromatid cohesion at least in part through antagonizing WAPL.
DOI: https://doi.org/10.7554/eLife.33920.013

ligation of Okazaki fragment and histone deposition (*Figure 4F*). These results are in agreement with a previous report in the budding yeast (*Borges et al., 2013*).

A recent in vitro reconstitution study with the fission yeast cohesin revealed that, in the presence of Scc2/4, the DNA-bound cohesin could topologically capture a second DNA molecule, which has to be single stranded DNA (ssDNA) (*Murayama et al., 2018*). Conversion of the captured ssDNA to dsDNA leads to a more stable cohesin-mediated DNA–DNA linkage. That study suggests that sister-chromatid cohesion is established at replication forks before lagging strand synthesis is completed, a finding in general agreement with ours.

## Replisome components are required for cohesin loading and MCM–NIPBL–cohesin interaction

We synchronized HeLa cells at different cell cycle stages with a nocodazole-arrest-release protocol (*Figure 5—figure supplement 1A*), isolated the chromatin fraction from these cells, and blotted the total lysates and chromatin fractions with antibodies against pre-RC, replisome, and cohesin components (*Figure 5—figure supplement 1B*). As expected, ORC2 remained bound to chromatin throughout the cell cycle. Loading of MCM2–7, CDC7, cohesin, and NIPBL/MAU2 began in telophase and reached peak levels at the G1/S boundary (6 hr after nocodazole release). Chromatin association of the replisome components CDC45, GINS1, WDHD1, and TIMELESS reached peak levels in S phase (6–10 hr after nocodazole release), concomitant with the chromatin association of ESCO2 and acetylated SMC3. These results confirm that the replisome components, particularly WDHD1 and TIMELESS, associate with chromatin mainly during S phase.

We next examined whether cohesin loading and the formation of the MCM–NIPBL–cohesin complex were impaired by depletion of replisome components. Depletion of WDHD1 or TIMELESS reduced cohesin loading in cells that were arrested in early S phase with thymidine (*Figure 5A–C*). Binding of NIPBL/MAU2 and cohesin to MCM2 was also reduced when WDHD1 or TIMELESS was depleted (*Figure 5D*). The binding was further reduced by the co-depletion of both. DDX11 depletion also led to reduced binding of NIPBL/MAU2 and cohesin to MCM2 (*Figure 5E*). Therefore, these replisome components are required for cohesin loading and the MCM–NIPBL–cohesin interaction.

Because these components only associate with the active CMG, not the pre-RC, their involvement in cohesin loading and the MCM–NIPBL–cohesin interaction strongly suggests that NIPBL/MAU2-dependent cohesin loading persists at active replication forks. We tested whether the replisome components were specifically required for cohesin loading and the binding of NIPBL/MAU2 and cohesin to the activated MCM2–7. As shown in *Figure 3—figure supplement 4*, depletion of CDC45, which was expected to inhibit MCM2–7 activation and enrich MCM2–7 in the pre-RC, did not reduce the binding of NIPBL/MAU2 and cohesin to MCM2. Co-depletion of CDC45 partially reversed the defect in NIPBL/MAU2 binding to MCM2 caused by TIMELESS depletion, but had no effect on that caused by CDC7 depletion (*Figure 5F*). Thus, our data suggest that CDC7 is required for the recruitment of NIPBL/MAU2 to the pre-RC whereas TIMELESS and other replisome

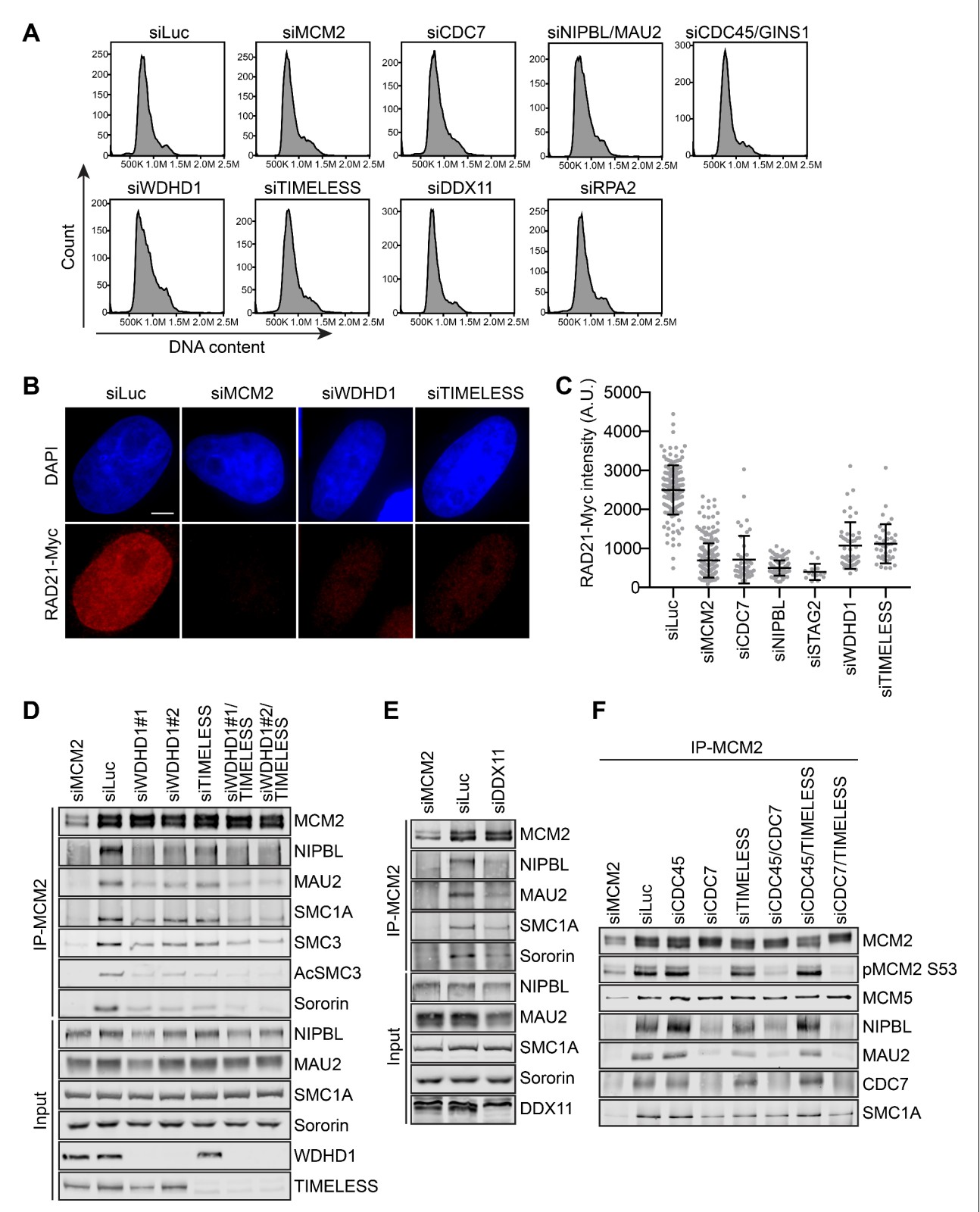

**Figure 5.** Replisome components are required for cohesin loading and the MCM–NIPBL–cohesin interaction during early S phase. (**A**) Flow cytometry analysis of HeLa cells transfected with the indicated siRNAs and treated with thymidine for 16–18 hr. The DNA content histograms were shown. (**B**) DAPI (blue) and anti-Myc (red) staining of HeLa cells that stably expressed RAD21-Myc. Cells were transfected with the indicated siRNAs and arrested in early S phase by thymidine before fixation and staining. Scale bar, 5 μm. (**C**) Quantification of the chromatin intensities of RAD21-Myc of cells in (**B**). Each dot

*Figure 5 continued on next page*

*Figure 5 continued*

in the graph represents a single cell. Mean ± SD (siLuc, n = 151; siMCM2, n = 238; siCDC7, n = 43; siNIPBL, n = 97; siSTAG2, n = 14; siWDHD1, n = 47; siTIMELESS, n = 39). (D–F) HeLa cells were transfected with the indicated siRNAs, arrested in early S phase by thymidine, and lysed in the presence of Turbo nuclease. The total lysates (input) and anti-MCM2 immunoprecipitate (IP) were blotted with the indicated antibodies.
DOI: https://doi.org/10.7554/eLife.33920.014
The following figure supplement is available for figure 5:

**Figure supplement 1.** Chromatin binding of replisome components peaks in S phase.
DOI: https://doi.org/10.7554/eLife.33920.015

components are only required for NIPBL/MAU2 recruitment to the active CMG helicase. Taken together, our results suggest that the MCM–NIPBL–cohesin interaction persists at active replication forks and requires replisome components for stabilization during DNA replication. Interestingly, cohesin is also recruited to active replication forks in the budding yeast, and that recruitment is crucial for the restart of stalled forks (*Tittel-Elmer et al., 2012*).

## RPA is required for the MCM–NIPBL–cohesin interaction and cohesion establishment

We next investigated the role of the RPA complex in sister-chromatid cohesion, which should be exclusively recruited to the active replication forks to stabilize ssDNA during DNA replication. RPA2-depleted cells displayed strong cohesion defect, as revealed by the FISH assay (*Figure 6A,B*). The cohesion defects in these cells was partially rescued by overexpression of RNAi-resistant Myc-RPA2 (*Figure 6C* and *Figure 6—figure supplement 1*). Similar results were obtained with a second FISH probe (465K16) that detected a late-replicating locus on chromosome 8 (*Figure 6—figure supplement 2*). Depletion of RPA or WDHD1/TIMELESS also caused cohesion defects in asynchronous cells without thymidine treatment (*Figure 6—figure supplement 3*), as assayed by both FISH probes, whereas depletion of FEN1 or LIG1 did not produce cohesion defects.

The level of cohesin on chromatin was significantly reduced in RPA2-depleted cells arrested at early S phase by thymidine treatment (*Figures 5A* and *6D,E and Figure 6—figure supplement 4*). Depletion of RPA2 reduced the binding of NIPBL/MAU2 and cohesin to MCM2 in early S phase (*Figure 6F*). The MCM–NIPBL–cohesin interaction could be partially restored in cells co-depleted of CDC45 and RPA2 (*Figure 6G*). Thus, RPA is required for cohesion establishment and for cohesin loading and the MCM–NIPBL–cohesin interaction during early S phase. Our findings further suggest that the simultaneous recruitment of NIPBL/MAU2 and cohesin to active replication forks is critical for sister-chromatid cohesion establishment during DNA replication.

Depletion RPA2, TIMELESS, or WDHD1 weakened CDC7 binding to MCM2 (*Figure 6F,G*). We next examined the phosphorylation status of MCM2 in cells depleted of replisome components and arrested at early S phase with thymidine. Depletion of RPA2 and other replisome components reduced the phosphorylation of MCM2 to varying degrees (*Figure 6H*). However, the reduction in MCM2 phosphorylation did not quantitatively correlate with the severity of cohesion defects caused by the depletion of each component. Finally, we could not detect RPA, WDHD1, TIMELESS, or DDX11 in MCM2 IPs from nuclease-treated lysates, suggesting that their interactions with the activated MCM2–7 helicase might be chromatin-dependent. These results suggest that RPA and other replisome components might directly contribute to the MCM–NIPBL–cohesin interaction on chromatin. They may also promote this interaction indirectly through regulating DDK.

## Discussion

### Requirement of MCM2–7 and DDK in cohesin loading

In *Xenopus* egg extracts, Scc2/4-dependent cohesin loading to chromatin depends on the pre-RC and DDK (*Gillespie and Hirano, 2004*; *Takahashi et al., 2008*; *Takahashi et al., 2004*). In addition, Scc2/4 exists in a stable complex with DDK, and the kinase activity of DDK is required to tether Scc2/4 and cohesin to the pre-RC and chromatin (*Takahashi et al., 2008*). We now show that, in human cells, cohesin loading during early S phase, but not telophase, requires the MCM2–7 complex, which is a core component of the pre-RC. MCM2–7 and DDK physically interact with NIPBL/

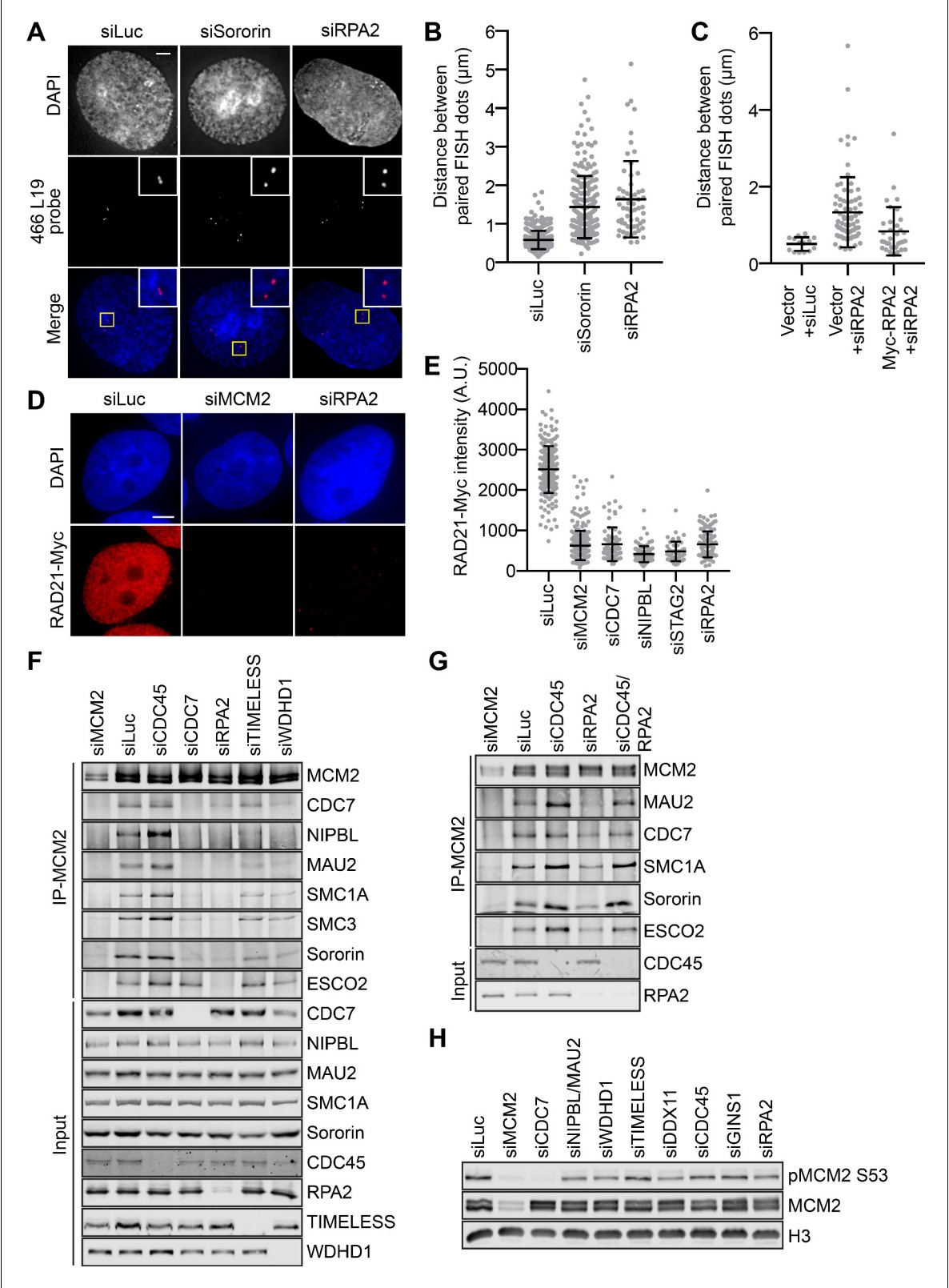

**Figure 6.** RPA promotes interphase sister-chromatid cohesion, cohesin loading in early S phase, and the MCM–NIPBL–cohesin interaction. (**A**) Representative images of G2-enriched HeLa cells transfected with the indicated siRNAs and stained with DAPI (blue in merge) and the early-replicating 466L19 FISH probe (red in merge). Selected paired FISH signals are magnified in inset. Scale bar, 5 μm. (**B**) Quantification of the distances between paired FISH signals of cells in (**A**). Mean ± SD (siLuc, n = 439; siSororin, n = 226; siRPA2, n = 59). (**C**) Quantification of the distances between paired FISH

*Figure 6 continued on next page*

*Figure 6 continued*

signals in G2-enriched HeLa cells transfected with the indicated plasmids and siRNAs. Mean ± SD (Vector + siLuc, n = 14; Vector + siRPA2, n = 78; Myc-RPA2 +siRPA2, n = 34). (D) DAPI (blue) and anti-Myc (red) staining of HeLa cells that stably expressed RAD21-Myc. Cells were transfected with the indicated siRNAs and arrested in early S phase with thymidine before fixation and staining. Scale bar, 5 μm. (E) Quantification of the chromatin intensities of RAD21-Myc of cells in (D). Mean ± SD (siLuc, n = 215; siMCM2, n = 298; siCDC7, n = 75; siNIPBL, n = 91; siSTAG2, n = 52; siRPA2, n = 131). (F,G) HeLa cells were transfected with the indicated siRNAs, synchronized in early S phase by thymidine, and lysed in the presence of Turbo nuclease. The total lysates (input) and anti-MCM2 immunoprecipitate (IP) were blotted with the indicated antibodies. (H) Lysates of HeLa cells transfected with the indicated siRNAs and arrested at early S phase with thymidine were blotted with the indicated antibodies.

DOI: https://doi.org/10.7554/eLife.33920.016

The following figure supplements are available for figure 6:

**Figure supplement 1.** RPA2 promotes sister-chromatid cohesion.

DOI: https://doi.org/10.7554/eLife.33920.017

**Figure supplement 2.** RPA2 promotes sister-chromatid cohesion.

DOI: https://doi.org/10.7554/eLife.33920.018

**Figure supplement 3.** Depletion of RPA causes interphase cohesion defects in asynchronous cells.

DOI: https://doi.org/10.7554/eLife.33920.019

**Figure supplement 4.** RPA2 promotes cohesin loading in S phase.

DOI: https://doi.org/10.7554/eLife.33920.020

MAU2 and cohesin. This interaction is regulated during the cell cycle and requires the kinase activity of DDK. Therefore, the function of MCM2–7 and DDK in cohesin loading is conserved in vertebrates. We propose that MCM2–7 and DDK interact with SCC2/4 and cohesin at the G1/S boundary and load cohesin to chromatin in the vicinity of the pre-RC. The loaded cohesin remains physically associated with SCC2/4 and MCM2–7 on chromatin (*Figure 7*). Our model is consistent with a recent study showing that SCC2 can still bind to chromatin primarily through an association with cohesin after the initial wave of cohesin loading in telophase and G1 (*Rhodes et al., 2017a*).

The dispensability of the MCM2–7-dependent mechanism for cohesin loading in telophase suggests the existence of MCM2–7-independent cohesin loading mechanisms in human somatic cells during telophase and G1. This possibility might explain why a published study did not detect a reduction of chromatin-bound cohesin in MCM2-depleted, asynchronous human cells (*Guillou et al., 2010*), as the majority of such cells were in G1. In *Drosophila*, ChIP-seq reveals considerable overlap between sites of ORC and cohesin binding throughout the genome (*MacAlpine et al., 2010*), suggesting that the pre-RC might be involved in cohesin loading in that organism. Depletion of the CDT1 ortholog in *Drosophila*, however, did not cause overt cohesin loading defects, arguing against that possibility (*MacAlpine et al., 2010*). It remains to be tested whether inactivation of the pre-RC selectively affects cohesin loading in early S phase in *Drosophila*. The *Xenopus* egg extract is a cell-free system that recapitulates the embryonic cell cycle, which lacks the G1 phase. Furthermore, there is no active transcription in the *Xenopus* egg extract. These differences provide possible explanations for the MCM2–7-dependent cohesin loading mechanism being predominant in that system. It will be interesting to test whether the mechanism of cohesin loading during telophase and G1 in human somatic cells involves loop extrusion and requires active transcription.

A further implication of our findings is that the MCM2–7-independent cohesin loading mechanism is attenuated in S phase. It is unclear how this is accomplished. An early study showed that the basal transcriptional activity is low in S phase, compared to early G1 (*Yonaha et al., 1995*). It is conceivable that transcription is globally attenuated during S phase to avoid collisions between the replication and transcription machineries. This attenuation leads to less efficient cohesin loading through the transcription-dependent G1 pathway. MCM-dependent cohesin loading then becomes the dominant pathway during early S phase. Future studies are needed to test this possibility.

In the budding yeast, the association of cohesin with chromatin throughout the cell cycle is not affected when the pre-RC assembly is inhibited by Cdc6 inactivation (*Uhlmann and Nasmyth, 1998*). The pre-RC is thus unlikely to be required for cohesin loading in yeast. Interestingly, the recruitment of Scc2/4 to yeast centromeres is dependent on the kinase activity of DDK during the G1/S transition (*Hinshaw et al., 2017*). Scc2/4 interacts directly with the Ctf19 kinetochore protein phosphorylated by DDK, and this DDK-dependent interaction is required for centromeric cohesion in yeast (*Hinshaw et al., 2017*). Thus, DDK-dependent cohesin loading might be conserved in all

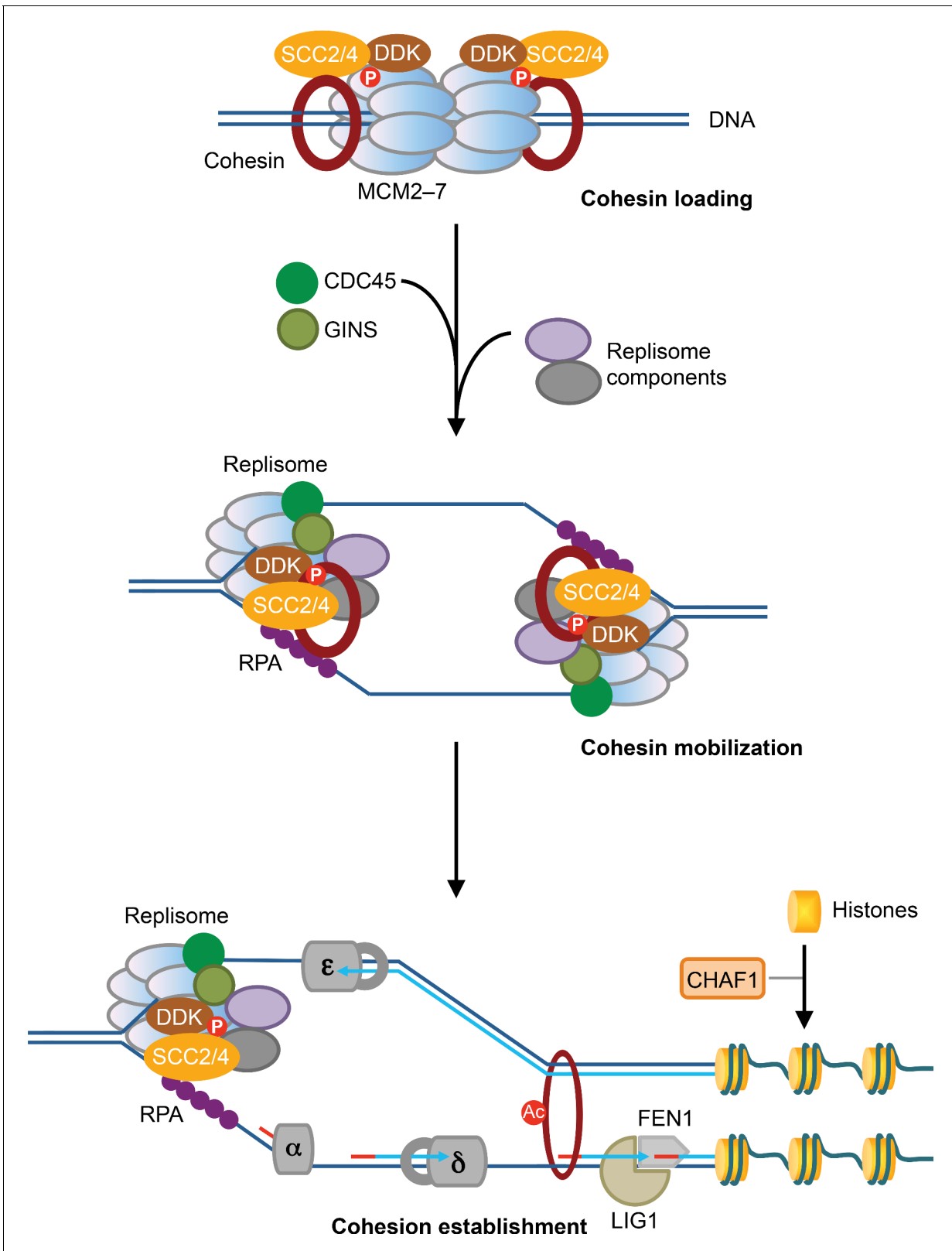

**Figure 7.** MCM2–7-mediated cohesin loading and mobilization promote sister-chromatid cohesion. In this speculative model, SCC2/4 (NIPBL/MAU2 in humans) associates with DDK and the dormant, phosphorylated MCM2–7, and promotes cohesin loading at the G1/S boundary. The loaded cohesin remains physically associated with SCC2/4, DDK, and MCM2–7. Upon the activation of the helicase activity of MCM2–7 and the initiation of DNA replication, cohesin bound to the dormant MCM2–7 is mobilized and held at the active replication forks, through a process that requires SCC2/4, DDK,

*Figure 7 continued on next page*

*Figure 7 continued*

and a multitude of replisome components, including RPA. Cohesin bound to the replication fork is then deposited behind the fork prior to the completion of lagging strand synthesis and histone deposition, and entraps both sister chromatids to establish sister-chromatid cohesion.

DOI: https://doi.org/10.7554/eLife.33920.021

The following figure supplement is available for figure 7:

**Figure 7—Figure Supplement 1.** DDK-bound, potentiated MCM promotes cohesin loading and mobilization.

DOI: https://doi.org/10.7554/eLife.33920.022

eukaryotes. Because Ctf19 is not conserved in vertebrates, SCC2/4 likely interacts with other DDK-mediated phospho-peptide motifs in MCM2–7 or DDK itself. Future studies are needed to define the detailed mechanisms of DDK-dependent recruitment of SCC2/4 and cohesin to MCM2–7.

## Mobilization of pre-loaded cohesin by SCC2/4 and replisome components

Our results and previously published reports clearly establish that the assembly of the active CMG helicase and the initiation of DNA replication are not required for MCM2–7-dependent cohesin loading at early S phase. Surprisingly, however, depletion of well-established replisome components, including WHDH1, TIMELESS, DDX11, and RPA, reduces the MCM–SCC2/4–cohesin interaction in thymidine-arrested early S phase cells. Preventing MCM2–7 activation by co-depletion of CDC45 partially alleviates the requirement for these replisome components in maintaining the MCM–SCC2/4–cohesin interaction.

It is widely accepted that there is an excess amount of MCM complexes loaded on chromosomes. Only a small pool of MCM complexes is activated to initiate and sustain DNA replication. DDK plays an important role in MCM activation. Using quantitative immunoblotting, we have determined that the molar ratio of total CDC7 (the catalytic subunit of DDK) to MCM2 is about 1:30 in HeLa cells. Thus, only 1 out of 30 MCM complexes can be bound by DDK at a given time. Our results suggest that only DDK-bound, potentiated MCM is capable of interacting with SCC2/4 and cohesin and supporting cohesin loading. This potentiated pool of MCM marked by DDK may be destined to fire first, converting the associated cohesin complexes into cohesive ones (*Figure 7—figure supplement 1*). After cohesin mobilization, DDK dissociates from these early origins and associates with MCM complexes at late-replicating origins. The processes of cohesin loading and mobilization then repeat at those origins. A large fraction of the pre-RC at early-replicating origins might have been activated to become active CMG in thymidine-arrested cells. SCC2/4 and cohesin remain associated with the active CMG, but this association now requires the actions of various replisome components. This additional requirement suggests that SCC2/4, cohesin, and DDK might interact with MCM2–7 in a different mode at active replication forks.

In WAPL-depleted human cells with little cohesin turnover, cohesin loaded prior to DNA replication can persist on chromatin throughout S phase (*Rhodes et al., 2017b*), suggesting that the DNA replication machinery does not necessarily displace cohesin already bound to un-replicated DNA. Although the diameter of the fully stretched cohesin ring is about 40 nm, the functional pore size of DNA-bound cohesin is less than 19.5 nm, and cohesin can be pushed along DNA by a motor protein with a diameter of 13 nm (*Stigler et al., 2016*). Being only a part of the DNA replication machinery, the CMG helicase has a cross-section of 11 nm in width and 17 nm in length, when viewed along the axis of DNA (*Georgescu et al., 2017*). It is challenging for the entire replication machinery to pass through the cohesin ring without the stretching or transient opening of the ring. Indeed, cohesin can be pushed along DNA by the replication machinery in *Xenopus* egg extracts (*Kanke et al., 2016*).

The simplest model to explain our results and the published reports is that cohesin loaded on un-replicated DNA and associated with the dormant MCM2–7 is mobilized upon MCM2-7 activation and transiently held at the active replication fork by SCC2/4, DDK, and other replisome components (*Figure 7*). Human MCM2 has been suggested to capture parental histones H3-H4 evicted by the replication fork and re-deposit them on the replicated DNA (*Huang et al., 2015*). Our proposed cohesin mobilization by MCM2–7 is conceptually similar to the mechanism of histone recycling by MCM2. This mobilization may involve the stretching of the cohesin ring or the opening and re-closure of the ring at one of its subunit interfaces.

## Cohesion establishment at the replication fork

Depletion of several replisome components reduces MCM2–7-dependent cohesin loading and the MCM–SCC2/4–cohesin interaction in cells arrested at early S phase, and causes interphase cohesion defects (*Figures 4* and *5*). These results suggest that the MCM–SCC2/4–cohesin interaction might be connected to cohesion establishment. We propose that cohesin loaded on un-replicated DNA is mobilized by SCC2/4 and held by the replication fork. SCC2/4 then promotes the capture of a second DNA molecule by the fork-bound cohesin, thus establishing sister-chromatid cohesion (*Figure 7*). Components required for Okazaki fragment maturation and ligation and histone deposition are dispensable for cohesion establishment, suggesting that cohesion establishment can occur prior to these events. Because RPA, the heterotrimeric single-stranded DNA-binding protein, is required for cohesion establishment, the capture of the second DNA molecule might involve single-stranded DNA and occur before the completion of lagging strand synthesis. This hypothesis is consistent with the recent in vitro study that demonstrates the Scc2/4-dependent capture of a second single-stranded DNA molecule by cohesin already bound to double-stranded DNA (*Murayama et al., 2018*).

Replisome components, including RPA, are required for proper DNA replication. Obviously, partial depletion of these components still permits DNA replication, minimally of the specific loci examined by FISH, thus revealing cohesion defect. On the other hand, we cannot exclude the possibility that aberrant DNA replication in cells deficient for these replisome components indirectly causes cohesion defects. Identification of separation-of-function mutants of replisome components that selectively disrupt cohesion establishment without perturbing DNA replication is needed to eliminate this caveat. Importantly, perturbation of DNA replication in and of itself is insufficient to trigger cohesion defect, as depletion of FEN1, LIG1, or CHAF1A interferes with DNA replication, but does not produce cohesion defect.

Depletion of WAPL rescues the cohesion defect caused by depletion of replisome components, suggesting that the replisome promotes cohesion establishment at least in part through antagonizing the anti-establishment activity of WAPL in human cells. WAPL needs to engage the scaffolding protein PDS5 to release cohesin from chromatin (*Ouyang and Yu, 2017*). SCC2/4 and PDS5 compete for cohesin binding and cannot simultaneously bind to cohesin (*Kikuchi et al., 2016*). Thus, cohesin bound to MCM–SCC2/4 is expected to be refractory to WAPL-dependent release. If replication-coupled, SCC2/4-dependent cohesin mobilization and deposition indeed allows the capture of the second DNA molecule, subsequent ESCO2-mediated SMC3 acetylation (also at the replication fork) may release SCC2/4 from cohesin, promote the binding of PDS5 and sororin to cohesin, and further stabilize sister-chromatid cohesion.

## Conclusion

We have established MCM2–7-dependent cohesin loading as a conserved mechanism in vertebrates and possibly other multicellular organisms. Cohesin and its loader SCC2/4 (NIPBL/MAU2 in humans) remain physically associated with the activated MCM2–7 in a process that requires the kinase activity of DDK and replisome components. We propose cohesin loading at the pre-RC and subsequent SCC2/4-dependent cohesin mobilization at replication forks promote the establishment of sister-chromatid cohesion.

## Materials and methods

### Mammalian cell culture, transfection, and synchronization

HeLa Tet-On cells were initially purchased from Clontech- (now Takara Bio USA, Inc., Mountain View, CA) and had been recently authenticated with STR profiling at ATCC (Manassas, VA). The cells were routinely monitored to make sure that they were free of mycoplasma contamination. HeLa Tet-On cells were grown in DMEM (Invitrogen, Carlsbad, CA) supplemented with 10% fetal bovine serum (FBS) and 2 mM L-glutamine. When cells reached a confluency of 50%, plasmid transfection was performed using the Effectene reagent (Qiagen) according to the manufacturer's protocols. All mammalian expression plasmids used in this study were derived from modified pCS2 vectors. The human MCM2, WDHD1, TIMELESS, and RPA2 cDNAs that contained silent mutations in the siRNA-targeted regions were inserted into these vectors that introduced HA or Myc tags at the N-terminus

of the coded proteins. All constructs were verified by DNA sequencing. The HeLa Tet-On cell line stably expressing RAD21-Myc was made as described previously (Wu et al., 2012). RAD21-Myc expression was induced with 1 µg/ml doxycycline. For siRNA transfection, cells at 20–40% confluency were transfected with Lipofectamine RNAiMAX (Invitrogen) according to the manufacturer's protocols, and analyzed at 24–48 hr after transfection. The siRNAs were transfected at a final concentration of 5 nM. The siRNAs used in this study were listed in *Supplementary file 1*.

For synchronization of cells at early S phase, cells were treated with 2 mM thymidine for 16–18 hr. For synchronization of cells at G2, cells were treated with the CDK1 inhibitor RO3306 at a final concentration of 10 µM for 18 hr. For immunofluorescence staining of cells in telophase, cells were treated with 2 mM thymidine for 16–18 hr and released into fresh medium containing 300 nM nocodazole (Sigma, Burlington, MA) for 12 hr to block cells at mitosis. Cells were then washed with PBS for three times and released into fresh medium for 4 hr. For inhibition of the kinase activity of CDC7, cells were treated with XL413 (Tocris Bioscience, Bristol, UK) at 5 µM for 6 hr.

## Antibodies

The following antibodies against human proteins were used for immunoblotting, immunofluorescence and immunoprecipitation: anti-NIPBL/SCC2 (Bethyl Laboratories, Montgomery, TX, A301-779A), anti-MAU2/SCC4 (Abcam, ab183033), anti-SMC1A (Bethyl Laboratories, A300-055A), anti-SMC3 (Bethyl Laboratories, A300-060A), anti-STAG2 (Santa Cruz Biotechnology, Dallas, TX, sc-81852), anti-RAD21/SCC1 (Bethyl Laboratories, A300-080A), anti-ESCO2 (Bethyl Laboratories, A301-689A), anti-MCM2 (Bethyl Laboratories, A300-191A), anti-MCM5 (Bethyl Laboratories, A300-195A), anti-MCM2 pSer53 (Bethyl Laboratories, A300-756A), anti-CDC7 (Bethyl Laboratories, A302-504A), anti-WDHD1 (Bethyl Laboratories, A301-141A), anti-TIMELESS (Bethyl Laboratories, A300-961A), anti-TIPIN (Bethyl Laboratories, A301-474A), anti-DDX11 (Santa Cruz Biotechnology, sc-68855), anti-CDC45 (Santa Cruz Biotechnology, sc-55569; Cell Signaling, D7G6), anti-GINS1 (Bethyl Laboratories, A304-170A), anti-RPA2 (Millipore, Burlington, MA, MABE285; Bethyl Laboratories, A300-244A), anti-ORC2 (Bethyl Laboratories, A302-734A), anti-CHAF1A (Bethyl Laboratories, A301-481A), anti-LIG1 (Abcam, Cambridge, MA, ab615), anti-FEN1 (Bethyl Laboratories, A300-255A), anti-Myc (Roche, Basel, Switzerland, 11667203001), anti-α-TUBULIN (Sigma, T9026; Bio-Rad, MCA77G), anti-Histone H3 (Abcam, ab1791). The anti-SMC3 K105Ac antibody was a gift from Dr. Prasad Jallepalli (Memorial Sloan Kettering Cancer Center). The anti-WAPL and anti-sororin antibodies were generated against recombinant human WAPL$_{601-1190}$ and sororin$_{91-252}$ proteins, respectively, as previously described (*Ouyang et al., 2013*; *Ouyang et al., 2016*).

## Immunoblotting and immunoprecipitation

For immunoblotting, cells were lysed in the SDS sample buffer (pH 6.8), sonicated, and boiled. The total lysates were separated by SDS-PAGE and blotted with the desired primary antibodies. The primary antibodies were used at a final concentration of 1 µg/ml. Anti-mouse IgG (H + L) (Dylight 680 conjugates) and anti-rabbit IgG (H + L) (Dylight 800 conjugates) (Cell Signaling, Danvers, MA) were used as secondary antibodies. The blots were scanned with an Odyssey Infrared Imaging System (LI-COR, Lincoln, NE) according to the manufacturer's protocols.

For immunoprecipitation, the anti-MCM2, anti-NIPBL, or anti-sororin antibodies were coupled to the Affi-Prep Protein A beads (Bio-Rad, Hercules, CA) at a concentration of 1 mg/ml. Cells were lysed with the lysis buffer containing 25 mM Tris-HCl (pH 7.7), 50 mM NaCl, 0.1% (v/v) Nonidet P-40, 2 mM MgCl$_2$, 10% (v/v) glycerol, 5 mM NaF, 0.3 mM Na$_3$VO$_4$, 10 mM β-glycerophosphate, 1 mM DTT, complete EDTA-free protease inhibitor cocktail (Roche), and 50 units/ml Turbo Nuclease (Accelagen, San Diego, CA). After a 1 hr incubation on ice and a 10 min incubation at 37°C, all lysates were centrifuged at 4°C at 20,817 g for 20 min. The supernatants were incubated with the desired antibody beads for 3 hr at 4°C. The beads were then washed three times with the lysis buffer. Proteins bound to beads were dissolved in SDS sample buffer, separated by SDS-PAGE, and blotted with the appropriate antibodies.

## Immunofluorescence

HeLa Tet-On cells were cultured and treated in the Nunc Lab-Tek II CC2 Chamber Slides. Cells on the slides were first permeabilized with the PHEM buffer (25 mM HEPES pH 7.5, 10 mM EGTA pH

8.0, 60 mM PIPES pH 7.0, and 2 mM MgCl$_2$) containing 0.5% Triton X-100 for 5 min and then fixed in 2% paraformaldehyde for 15 min. Fixed cells were blocked in PBS containing 2% BSA for 30 min and then incubated with desired antibodies in PBS containing 0.1% Triton X-100 (PBST) and 3% BSA and at 4°C overnight. Cells were then washed three times with PBST for 5 min each time, and incubated with fluorescent secondary antibodies (Molecular Probes) in PBST containing 3% BSA for 1 hr at room temperature. Cells were again washed three times with PBST and stained with 1 µg/ml DAPI in PBS for 5 min. After the final wash with PBS, the slides were mounted with VECTASHIELD antifade mounting medium (Vector Laboratories), sealed with nail polish, and viewed with a 100X objective on a DeltaVision fluorescence microscope (GE Healthcare). Image processing and quantification were performed with Image J.

## Chromatin fractionation

Cells were harvested by trypsinization and washed once with PBS. Cells were then resuspended with ice-cold fractionation buffer containing 25 mM Tris-HCl pH 7.7, 50 mM NaCl, 0.1% (v/v) Nonidet P-40, 2 mM MgCl$_2$, 10% (v/v) glycerol, 5 mM NaF, 0.3 mM Na$_3$VO$_4$, 10 mM β-glycerophosphate, 1 mM DTT, complete EDTA-free protease inhibitor cocktail (Roche), and 5 mM sodium butyrate. The cell suspension was passed through a 27G × 1/2 inch (0.4 × 13 mm) needle 7–10 times, and incubated on ice for 10 min. Cell lysates were then centrifuged at 4°C at 3,000 g for 5 min. The pellet was gently resuspended and washed with the fractionation buffer for three times. The chromatin fractions were then lysed in SDS sample buffer, sonicated, boiled, separated by SDS-PAGE, and blotted with the desired antibodies.

## Fluorescence in situ hybridization (FISH)

The BAC clones RP11-466L19 and RP11-465K16 were purchased from Empire Genomics. FISH probes were labeled with 5-Fluorescein dUTP (Enzo Life Sciences, Farmingdale, NY) using the Nick Translation Kit (Abbott Molecular, Des Plaines, IL). Probes were precipitated with human cot-1 DNA (Invitrogen) and salmon sperm DNA (Invitrogen), and then resuspended in the hybridization solution (Cytocell, Tarrytown, NY). HeLa Tet-on cells were transfected with siRNAs, synchronized with 2 mM thymidine for 16–18 hr and then released into fresh medium for 4 hr. Cells were harvested by trypsinization, treated with 75 mM KCl hypotonic solution for 25 min at 37°C, and then fixed with ice-cold methanol and acetic acid (ratio 3:1). Fixed cells were dropped onto pre-warmed slides, in situ hybridized at 80°C with DNA probes and incubated at 37°C overnight. Slides were sequentially washed with 0.1% SDS in 0.5 X SSC at 70°C for 5 min, 1 X PBS at room temperature for 10 min and 0.1% Tween 20 in PBS at room temperature for 10 min. Slides were then mounted with ProLong Gold (Life Technologies, Waltham, MA) and viewed with a 100X objective on a DeltaVision fluorescence microscope (GE Healthcare, Chicago, IL). Image processing and quantification were performed with ImageJ.

For the RP11-466L19 probe, only the cells with two pairs of FISH signals were processed, and the distances between paired FISH signals were quantified. The aneuploid HeLa cells contains three copies of the chromosome locus recognized by the RP11-465K16 probe. For that probe, cells with two or three pairs of FISH signals were processed, and the distances between paired FISH dots were quantified. In cells with strong cohesion defects, the two closest FISH dots were assumed to be a pair. Cells were excluded from the analysis if the ratio of the two distances between the two individual pairs of dots exceeded three-fold.

## Chromatin immunoprecipitation followed by next-generation sequencing (ChIP-seq)

HeLa Tet-On cells were synchronized with 2 mM thymidine for 16–18 hr. Cells from two 100 mm plates at full confluency were collected for each ChIP experiment. Cells in the plate were chemically crosslinked by the addition of 1% formaldehyde for 10 min followed with 125 mM glycine for 5 min at room temperature with shaking. Cells were then rinsed twice with ice-cold PBS and scraped from the plate into PBS containing 2X protease inhibitors. Cells were centrifuged at 4°C at 500 g for 5 min and washed one more time with PBS. Cells were lysed in the sonication buffer containing 10 mM Tris-HCl pH 7.4, 1 mM EDTA pH 8.0, 0.1% SDS, 1% Triton X-100, 0.1% sodium deoxycholate, 0.25% sarkosyl, 1 mM DTT, and protease inhibitor cocktail (Roche), and sonicated on ice to solubilize and

shear the crosslinked DNA to 300–500 bp. The resulting whole cell extract was centrifuged at 4°C at 17,949 g for 10 min. The supernatant was transferred to a new tube, and incubated with the appropriate antibody at 4°C overnight. For RAD21- and NIPBL-occupied genomic regions, anti-RAD21 ChIP Grade (Abcam, ab992) and anti-NIPBL/SCC2 (Bethyl Laboratories, A301-779A) antibodies were used for ChIP-seq experiments, respectively. Each sample was then incubated with pre-washed Dynabeads Protein A or G (Invitrogen) magnetic beads at 4°C for 3 hr. Beads were washed twice with the sonication buffer, twice with the sonication buffer containing 0.3 M NaCl, twice with the LiCl buffer (10 mM Tris-HCl pH 8.1, 1 mM EDTA pH 8.0, 250 mM LiCl, 0.5% (v/v) Nonidet P-40, 0.5% sodium deoxycholate), and twice with the TE buffer (10 mM Tris-HCl pH 7.5 and 1 mM EDTA pH 8.0). Bound complexes were eluted from the beads, and crosslinking was reversed by overnight incubation at 65°C in the SDS elution buffer (50 mM Tris-HCl pH 8.1, 1 mM EDTA pH 8.0, and 1% SDS). Immunoprecipitated DNA was treated with RNase A (Qiagen) and Proteinase K (New England Biolabs), and then purified with the PCR purification Kit (Qiagen) according to the manufacturer's protocols.

For ChIP-seq analysis, single-end reads of 75 bp were generated. After mapping the reads to the human genome (hg19) by bowtie2 (v2.2.3) with the parameter '–sensitive' (*Langmead and Salzberg, 2012*), we performed filtering by first removing alignments with mapping quality less than 10, and then removing duplicate reads identified by Picard MarkDuplicates (v1.127) (http://broadinstitute. github.io/picard). The enriched regions (peaks) were identified using MACS2 (v2.0.10) (*Zhang et al., 2008b*), with a q-value cut-off of 0.05 for broad peaks. Peak regions were annotated by HOMER (*Ross-Innes et al., 2012*). Co-localization plots were made using deepTools (v1.6) (*Ramírez et al., 2016*). SICER was used for peak calling with the MCM2 sample (*Xu et al., 2014*). ChIPseeker was used to make the annotation and co-occupancy plots for MCM2, RAD21, and NIPBL (*Yu et al., 2015*). All ChIP-seq datasets generated in this study have been deposited in GEO under the accession number GSE112028.

## Flow cytometry and BrdU incorporation

Cells were harvested with trypsinization and fixed in 70% ice-cold ethanol overnight. After being washed with PBS once, cells were resuspended in PBS containing 0.1% Triton X-100, 20 μg/ml propidium iodide (Sigma), and 200 μg/ml RNase A (Qiagen, Hilden, Germany), and incubated at room temperature for 1 hr. The samples were analyzed on a FACSCalibur flow cytometer (BD Biosciences). Data were processed with the FlowJo software.

For the BrdU incorporation assay, cells were incubated with 10 μM BrdU for 1 hr and harvested with trypsinization and fixed in 70% ice-cold ethanol overnight. After being washed with PBS once, cells were incubated with 3N HCl at room temperature for 30 min. Cells were washed twice with the phosphate/citric acid buffer (the mixture of 40 ml of 0.2 M $Na_2HPO_4$ and 3.956 ml of 0.1 M citric acid, pH 7.4) and once with PBS containing 1% BSA and 0.1% Triton X-100. Cells were then incubated with the FITC conjugated anti-BrdU antibody (BD Biosciences, 556028) at room temperature for 1 hr. Cells were then processed for flow cytometry analysis as described above.

## Acknowledgements

We thank Dr. Johannes Walter for helpful discussions and Dr. Prasad Jallepalli for providing the acetylated SMC3 antibody. This study is supported by grants from the Cancer Prevention and Research Institute of Texas (RP120717-P2 and RP160667-P2) and the Welch Foundation (I-1441). HY is an Investigator with the Howard Hughes Medical Institute. CX was supported in part by the National Institutes of Health grant UL1TR001105.

## Additional information

### Funding

| Funder | Grant reference number | Author |
| --- | --- | --- |
| Howard Hughes Medical Institute | | Hongtao Yu |

| | | |
|---|---|---|
| Welch Foundation | I-1441 | Hongtao Yu |
| Cancer Prevention and Research Institute of Texas | RP120717-P2 | Hongtao Yu |
| Cancer Prevention and Research Institute of Texas | RP160667-P2 | Hongtao Yu |

The funders had no role in study design, data collection and interpretation, or the decision to submit the work for publication.

## Author contributions

Ge Zheng, Conceptualization, Data curation, Formal analysis, Investigation, Visualization, Methodology, Writing—original draft; Mohammed Kanchwala, Software, Formal analysis, Visualization; Chao Xing, Data curation, Software, Formal analysis, Supervision; Hongtao Yu, Conceptualization, Supervision, Funding acquisition, Project administration, Writing—review and editing

## Author ORCIDs

Chao Xing (iD) http://orcid.org/0000-0002-1838-0502
Hongtao Yu (iD) http://orcid.org/0000-0002-8861-049X

## Decision letter and Author response

Decision letter https://doi.org/10.7554/eLife.33920.028
Author response https://doi.org/10.7554/eLife.33920.029

# Additional files

## Supplementary files

• Supplementary file 1. siRNAs used in this study.
DOI: https://doi.org/10.7554/eLife.33920.023

• Transparent reporting form
DOI: https://doi.org/10.7554/eLife.33920.024

## Major datasets

The following dataset was generated:

| Author(s) | Year | Dataset title | Dataset URL | Database, license, and accessibility information |
|---|---|---|---|---|
| Zheng G, Kanchwala M, Xing C, Yu H | 2018 | MCM2-7-dependent cohesin loading during S phase promotes sister-chromatid cohesion | https://www.ncbi.nlm.nih.gov/geo/query/acc.cgi?acc=GSE112028 | Publicly available at the NCBI Gene Expression Omnibus (accession no. GSE112028) |

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
