## [Decision Letter]

Thank you for submitting your article "MCM2-7-dependent cohesin loading during S phasepromotes sister-chromatid cohesion" for consideration by *eLife*. Your article has been reviewed by two peer reviewers, and the evaluation has been overseen by a Reviewing Editor and Andrea Musacchio as the Senior Editor. The reviewers have opted to remain anonymous.

The reviewers have discussed the reviews with one another and the Reviewing Editor has drafted this decision to help you prepare a revised submission.

Summary:

This manuscript centers on the role of the replisome in establishing sister chromatid cohesion during DNA replication. In a human cell line, the authors demonstrate that the replisomal factors MCM2-7 and DDK are critical in loading cohesin on the DNA during early S phase and a number of replisome-associated proteins are involved in retention of cohesion to DNA. The work reconciles a number of previous observation and establishes a detailed model for cohesion establishment in metazoans.

Essential revisions:

Subsection “Cohesin loading requires MCM2–7 in human cells during early S phase”, fourth paragraph, Figure 2B, C. The authors claimed that they determined the genome-wide distribution of MCM2, SCC2 and SCC1. However, the reviewer was not able to find data regarding the ChIP-seq of MCM2. In addition, the authors found that most of the SCC1 peaks were co-occupied by CTCF. Again, no data was reported on this and it was not clear whether they performed the genome-wide distribution of CTFC as well or if they used published data. Finally, was NGS data deposited in a public database?

Subsection “Replisome components promote the establishment of sister-chromatid cohesion”, first paragraph, Figure 4A. In order to examine cohesion status, the authors performed a FISH assay using a probe that specifically recognized a locus on chromosome 3. No information was provided on the replication status of that locus. In fact, it is well-known that almost 50% of genome is replicated early in S phase and the remaining 50% is replicated late. FISH experiments should be performed using two different probes (replicated early and late) at least in both unsynchronized and synchronized cells.

Other points:

1). The Introduction provides a succinct overview of a fast moving field. It is therefore important to cite current reviews of the field, several have recently been published, rather than ones that date back as far as 2002.

2) The Introduction surveys how cohesin is stabilized on DNA against unloading by WAPL. Three references from the authors' own lab are included, Ouyang et al., 2013, 2016 and 2017. While Ouyang et al., 2013 reported the structure of WAPL together with a proposal how WAPL opens the cohesin ring, Lopez-Serra et al. in the same year provided the first direct evidence for cohesin stabilization from WAPL by acetylation, which could be cited alongside.

3) 'Initial experiments in yeast showed that inactivation of the Scc2/4 complex in cells enriched in early S phase by hydroxyurea did not cause cohesion defect,… (Lengronne 2006).' That study was concerned with cell viability rather than a specific cohesion defect. Scc2 is not required to retain cell viability during S phase, though based on recent results from the Nasmyth and Uhlmann labs, it seems clear that cells develop a cohesion defect if Scc2 lacks in S phase that is compatible with viability. This could be rephrased.

4) Figure 1 shows MCM-depleted cells, arrested in early S phase by thymidine treatment. How is this possible? S phase arrest requires that replication initiates at least at some origins, so as to trigger the replication checkpoint. Following MCM depletion, the replication checkpoint would be inactive and cell cycle arrest no longer possible. Have potential cell cycle differences in this experiment been taken into account?

5) An important implication from this work is that cohesin maintenance on chromosomes in telophase and G1 is MCM-independent, but that it turns MCM-dependent during S phase. For cohesin to be lost from DNA during S phase, the telophase-G1 loading pathway must switch off. What might the basis for this be? Can the authors explore this in their Discussion?

6) Lower MCM levels during S phase lead to cohesin loss during DNA replication. However, the replisome must contain an MCM helicase for replication to proceed. Are the authors suggesting that MCM levels on top and above those required for DNA replication contribute to cohesin retention? This is a rather different interpretation of these results and this should be discussed.

7) 'We next determined the genome wide distribution of MCM2, SCC2/4 and cohesin…' This could be the beginning of a new paragraph.

8) The presentation of the ChIP experiment in Figure 2C could be improved. Is all the data centred on the Scc2 peaks? In this case, why is the average Scc2 peak, shown on top, off centre? How many Scc2 peaks are presented in both panels?

9) The Scc2 binding pattern in higher eukaryotes is still incompletely understood. The authors note that 'The majority of these SCC2 peaks were co-occupied by cohesin in two separate experiments, consistent with the notion that SCC2 remained bound to cohesin on chromatin'. Alternatively, Scc2 could demarcate cohesin loading sites to which cohesin is transiently recruited. It has been suggested that Scc2 occupies promoters (Kagey 2010, Lopez-Serra 2014, Zuin et al., 2014). Can the authors confirm or refute this based on their dataset? Can they say more about the detected Scc2 peaks?

10) Tittel-Elmer et al., 2012 found cohesin to colocalise with advancing DNA replication forks. If a similar assay is not possible to perform in human cells, this study could be cited.

11) Figure 3 and especially Figure 3—figure supplement 2B reports that phosphorylated MCM2 interacts with Scc2. This is based on the interaction with the faster migrating MCM2 form. This faster migration behaviour occurs rarely in case of phosphorylated proteins, so it would be good to confirm the band identities by phosphatase treatment.

12) Figure 3—figure supplement 4. The observation that GINS depletion does not affect cohesin loading is interesting. These are negative results. If the authors have a western blot or other functional confirmation for CDC45 or GINS depletion, this would be a suitable addition to this Figure.

13) The epistasis analysis of cohesion establishment factors in Figure 4B is confounded by potentially partial protein depletion by RNAi. Note that Scc2 depletion gives a relatively modest cohesion defect, which would be expected to result in no cohesion at all. RNAi of multiple components often results in further reduced depletion efficiency. Any conclusions about synthetic interactions, or lack thereof, should therefore be made cautiously. This also applies to the rescue of cohesion defects by WAPL depletion (Figure 4—figure supplement 1).

14) The interesting observation that FEN1 and nucleosome assembly components have no detectable role in sister chromatid cohesion establishment is in line with results in budding yeast (Borges et al., 2013), which could be mentioned.

15) The authors suggest that RPA and other replisome components contribute to the MCM-SCC2/4-cohesin interaction indirectly through regulating DDK. This is an interesting result that goes towards the mechanism of cohesin-replisome interactions. To substantiate this point, it would be interesting to quantitatively compare CDC7 loss from MCM2 with cohesin loss from chromosomes and sister chromatid cohesion defects following depletion of different replisome components. Do these measures correlate and follow a quantitatively similar pattern?

16) Recently, it has been shown that MCM complex and cohesin co-IPed and shared a number of sites in both unsynchronized and synchronized cells. Furthermore, they are enriched at replication origins (Cucco et al., 2017 Nucleic Acids Res. Nov 20. doi: 10.1093/nar/gkx1172). These data should be discussed.

17) Figure 2. Inputs of CDC7 and MCM2 are lacking.

18) Figures 3B and 3F are confusing. They show the same data (with the exception of XL413). Can the authors arrange them into a single figure?

19) Figure 3F. Inputs of ESCO, SA2, SCC4, pMCM2 and MCM5 are lacking.

20) Since somatic cells were used, SMC1 should be SMC1A.

21) Gene nomenclature should be referred as human.

---

## [Author Response]

Essential revisions:Subsection “Cohesin loading requires MCM2–7 in human cells during early S phase”, fourth paragraph, Figure 2B, C. The authors claimed that they determined the genome-wide distribution of MCM2, SCC2 and SCC1. However, the reviewer was not able to find data regarding the ChIP-seq of MCM2. In addition, the authors found that most of the SCC1 peaks were co-occupied by CTCF. Again, no data was reported on this and it was not clear whether they performed the genome-wide distribution of CTFC as well or if they used published data. Finally, was NGS data deposited in a public database?

We have now more fully described the ChIP-seq data on MCM2. We did not observe many MCM2 peaks using MACS2, perhaps due to the rather broad localization of the MCM complex in early S phase cells. In the revised manuscript, we used a clustering approach called SICER to identify of MCM2 enriched domains. We could identify many such MCM2-enriched domains. When NIPBL (SCC2) peaks were used as the reference, these MCM2 domains and SCC1 peaks were found to co-localize with 1,041 NIPBL peaks. We did not perform ChIP-seq on CTCF. Homer de novo motif analysis revealed that the CTCF motif was the top DNA consensus sequence for RAD21 (SCC1) binding, with scores of 0.915 and 0.914 in two independent experiments. In addition, we can readily observe co-localization of RAD21 with CTCF in the human genome using the CTCF ChIP-seq data in the ENCODE database. These findings confirm that cohesin is enriched at CTCF sites. Because CTCF does not impair sister-chromatid cohesion, we reason that the pool of cohesin enriched at CTCF sites might be different from the pool that associates with NIPBL and MCM. These new analyses are included in Figure 2—figure supplement 1. The ChIP-seq data have been deposited into GEO (https://www.ncbi.nlm.nih.gov/geo/query/acc.cgi?acc=GSE112028).

Subsection “Replisome components promote the establishment of sister-chromatid cohesion”, first paragraph, Figure 4A. In order to examine cohesion status, the authors performed a FISH assay using a probe that specifically recognized a locus on chromosome 3. No information was provided on the replication status of that locus. In fact, it is well-known that almost 50% of genome is replicated early in S phase and the remaining 50% is replicated late. FISH experiments should be performed using two different probes (replicated early and late) at least in both unsynchronized and synchronized cells.

We thank the reviewers for this great suggestion. Our 466L19 FISH probe included in the original submission recognizes the genomic region of Chr3, p24.3, 19050203-19251993. We have now included another 465K16 FISH probe that binds to the genomic region of Chr8, p11.21, 40659701-40853404. We first performed FISH assays with these probes in cells that were released from thymidine arrest at different times (Figure 6—figure supplement 2A). As cells progress through S phase, the percentage of cells with one or more pairs of FISH dots increases, even though the replication of the homologues is not completely synchronous in our assay. In HeLa cells with fully replicated loci, two pairs of dots could be detected with the 466L19 FISH probe, and three pairs of dots could be detected with the 465K16 probe. (Note that HeLa cells are aneuploid. They presumably contain three copies of the genomic locus on chromosome 8 recognized by the 465K16 probe.) Based on this assay, the genomic locus recognized by the 466L19 probe was replicated earlier as compared to that recognized by the 465K16 probe.

We observed similar cohesion defects in G2 cells depleted of Sororin,

WDHD1/TIMELESS, or RPA2 with the 465K16 probe that detects the late-replicating locus (Figure 6—figure supplement 2B, C). Likewise, depletion of FEN1 or LIG1 did not impair cohesion at the late-replicating locus recognized by the 465K16 probe. Additionally, we performed FISH experiments in asynchronized cells with both probes (Figure 6—figure supplement 3). We observed an increased number of cells that had completed replication in both genomic loci when Sororin, WDHD1/TIMELESS, or RPA2 was depleted in the absence of thymidine synchronization, consistent with the notion that depletion of these replication factors causes DNA damage and G2 delay under these conditions. For the 466L19 probe, only the cells with two pairs of FISH dots were analyzed, and the distances between paired FISH dots were quantified. For the 465K16 probe, only cells with two or three pairs of FISH dots were analyzed, and the distances between paired FISH dots were quantified. As expected, we observed cohesion defects in cells depleted of Sororin, WDHD1/TIMELESS, or RPA2, but not in cells depleted of FEN1 or LIG1 using both FISH probes. These new data with an additional FISH probe and in cells without synchronization are entirely consistent with and greatly strengthen our original conclusions.

Other points:1). The Introduction provides a succinct overview of a fast moving field. It is therefore important to cite current reviews of the field, several have recently been published, rather than ones that date back as far as 2002.

We have now referenced more current reviews of the cohesin field.

2) The Introduction surveys how cohesin is stabilized on DNA against unloading by WAPL. Three references from the authors' own lab are included, Ouyang et al., 2013, 2016 and 2017. While Ouyang et al., 2013 reported the structure of WAPL together with a proposal how WAPL opens the cohesin ring, Lopez-Serra et al. in the same year provided the first direct evidence for cohesin stabilization from WAPL by acetylation, which could be cited alongside.

We apologize for this oversight. The Lopez-Serra et al. study has now been referenced.

3) 'Initial experiments in yeast showed that inactivation of the Scc2/4 complex in cells enriched in early S phase by hydroxyurea did not cause cohesion defect,… (Lengronne 2006).' That study was concerned with cell viability rather than a specific cohesion defect. Scc2 is not required to retain cell viability during S phase, though based on recent results from the Nasmyth and Uhlmann labs, it seems clear that cells develop a cohesion defect if Scc2 lacks in S phase that is compatible with viability. This could be rephrased.

We have rephrased this statement to more accurately describe the findings of the Lengronne study. We note, however, that the Lengronne study did specifically examine cohesion status. In that paper (on p789), the authors stated that “We further confirmed that sister chromatid cohesion was intact in scc2-4 cells that traversed S phase at restrictive temperature (8% cells with separated GFP-marked URA3 loci in G2/M, compared to 7% after release at permissive temperature).”

4) Figure 1 shows MCM-depleted cells, arrested in early S phase by thymidine treatment. How is this possible? S phase arrest requires that replication initiates at least at some origins, so as to trigger the replication checkpoint. Following MCM depletion, the replication checkpoint would be inactive and cell cycle arrest no longer possible. Have potential cell cycle differences in this experiment been taken into account?

Depletion of the MCM complex in human cells did not apparently affect the early S phase arrest exerted by thymidine. As shown in Figure 5A, cells depleted of MCM2 treated with thymidine had a very similar cell cycle profile as compared to control depleted cells treated with thymidine. In the absence of thymidine, cells depleted of MCM2 for 48 hours ultimately arrested at G2. Thus, the depletion of MCM2 is partial, and does not completely block DNA replication. As we discussed, this partial depletion allows DNA replication to proceed (albeit less efficiently), thus revealing cohesion defects at the improperly replicated loci. To minimize complications from cell cycle differences, we performed cohesin immunostaining and immunoprecipitation under thymidine treatment conditions, which arrested cells depleted of various replication factors in early S phase (see Figure 5A).

5) An important implication from this work is that cohesin maintenance on chromosomes in telophase and G1 is MCM-independent, but that it turns MCM-dependent during S phase. For cohesin to be lost from DNA during S phase, the telophase-G1 loading pathway must switch off. What might the basis for this be? Can the authors explore this in their Discussion?

This is a great suggestion. We do not know the nature of the telophase/G1 pathway of cohesin loading and the mechanism by which this pathway is inactivated during S phase. One possibility is that general transcription facilitates cohesin loading onto chromosomes in telophase and early G1, possibly through loop extrusion. An early study showed that the basal transcription activity was low in S phase, compared to early G1 (Yonaha et al., Nucleic Acids Res., 1995). It is conceivable that transcription is globally attenuated during S phase to avoid collisions between the replication and transcription machineries. This attenuation leads to less efficient cohesin loading through the transcription-dependent pathway. MCM-dependent cohesin loading then becomes the dominant pathway during early S phase. Future studies are needed to test these possibilities. We have discussed this point in the revised manuscript.

6) Lower MCM levels during S phase lead to cohesin loss during DNA replication. However, the replisome must contain an MCM helicase for replication to proceed. Are the authors suggesting that MCM levels on top and above those required for DNA replication contribute to cohesin retention? This is a rather different interpretation of these results and this should be discussed.

It is widely accepted that there is an excess amount of MCM complexes loaded on chromosomes. Only a small pool of MCM complexes is activated to initiate and sustain DNA replication. DDK plays an important role in MCM activation. Using quantitative immunoblotting, we have determined that the molar ratio of total CDC7 (the catalytic subunit of DDK) to MCM2 is about 1:30 in HeLa cells. Thus, only 1 out of 30 MCM complexes can be bound by DDK at a given time. Our results suggest that only DDK-bound, potentiated MCM is capable of interacting with NIPBL and cohesin and supporting cohesin loading. This potentiated pool of MCM marked by DDK may be destined to fire first, converting the associated cohesin complexes into cohesive ones. DDK then associates with other MCM complexes at late-replicating origins. The processes of cohesin loading and mobilization then repeat at those origins. We have added a figure (Figure 7—figure supplement 1) and discussion to further clarify this point.

7) 'We next determined the genome wide distribution of MCM2, SCC2/4 and cohesin…' This could be the beginning of a new paragraph.

This is a great suggestion. We have modified the text as suggested.

8) The presentation of the ChIP experiment in Figure 2C could be improved. Is all the data centred on the Scc2 peaks? In this case, why is the average Scc2 peak, shown on top, off centre? How many Scc2 peaks are presented in both panels?

We thank the reviewer for catching this error. The figures in the original submission were actually made with the start of the SCC2 peaks aligned. We have now revised the figure with the center of the SCC2 peaks aligned. All SCC2 peaks (1,041 in Experiment 1 and 632 in Experiment 2) are included in the analysis.

9) The Scc2 binding pattern in higher eukaryotes is still incompletely understood. The authors note that 'The majority of these SCC2 peaks were co-occupied by cohesin in two separate experiments, consistent with the notion that SCC2 remained bound to cohesin on chromatin'. Alternatively, Scc2 could demarcate cohesin loading sites to which cohesin is transiently recruited. It has been suggested that Scc2 occupies promoters (Kagey 2010, Lopez-Serra 2014, Zuin et al., 2014). Can the authors confirm or refute this based on their dataset? Can they say more about the detected Scc2 peaks?

We analyzed the ChIP-seq data on NIPBL and annotated the genome-wide location of the NIPBL peaks in terms of genomic features. In both experiments, over 50% of NIPBL peaks were found at promoter regions. Our findings are thus in general agreement with the previous reports. We have included these data in Figure 2—figure supplement 2.

10) Tittel-Elmer et al., 2012 found cohesin to colocalise with advancing DNA replication forks. If a similar assay is not possible to perform in human cells, this study could be cited.

We have cited and discussed the Tittel-Elmer study.

11) Figure 3 and especially Figure 3—figure supplement 2B reports that phosphorylated MCM2 interacts with Scc2. This is based on the interaction with the faster migrating MCM2 form. This faster migration behaviour occurs rarely in case of phosphorylated proteins, so it would be good to confirm the band identities by phosphatase treatment.

To confirm the faster migrating band is the phosphorylated MCM2, we treated the HeLa cell lysates with λPPase in vitro. The faster migrating band disappeared after λPPase treatment, confirming that the faster migrating band indeed belonged to phosphorylated MCM2. These results are now included in Figure 3C.

12) Figure 3—figure supplement 4. The observation that GINS depletion does not affect cohesin loading is interesting. These are negative results. If the authors have a western blot or other functional confirmation for CDC45 or GINS depletion, this would be a suitable addition to this Figure.

We performed immunoblotting to confirm that CDC45 or GINS1 protein levels were greatly reduced after depletion. Additionally, we observed an increased number of cells accumulated in S phase when either CDC45 or GINS1 was depleted, as revealed by the BrdU incorporation assay in asynchronized cells. The cells in S phase did not incorporate BrdU as efficiently as the control cells. These results indicate that depletion of CDC45 or GINS1 is efficient enough to slow S phase progression. We have added these data to this figure. Obviously, the depletion is still incomplete, as complete depletion is expected to complete block S phase progression.

13) The epistasis analysis of cohesion establishment factors in Figure 4B is confounded by potentially partial protein depletion by RNAi. Note that Scc2 depletion gives a relatively modest cohesion defect, which would be expected to result in no cohesion at all. RNAi of multiple components often results in further reduced depletion efficiency. Any conclusions about synthetic interactions, or lack thereof, should therefore be made cautiously. This also applies to the rescue of cohesion defects by WAPL depletion (Figure 4—figure supplement 1).

We agree entirely with the reviewer about this point. We have toned down our conclusions about synthetic interactions and epistasis, and mentioned the caveat of partial depletions of various replisome components and WAPL.

14) The interesting observation that FEN1 and nucleosome assembly components have no detectable role in sister chromatid cohesion establishment is in line with results in budding yeast (Borges et al., 2013), which could be mentioned.

We have now cited the Borges et al. study and mentioned the budding yeast data.

15) The authors suggest that RPA and other replisome components contribute to the MCM-SCC2/4-cohesin interaction indirectly through regulating DDK. This is an interesting result that goes towards the mechanism of cohesin-replisome interactions. To substantiate this point, it would be interesting to quantitatively compare CDC7 loss from MCM2 with cohesin loss from chromosomes and sister chromatid cohesion defects following depletion of different replisome components. Do these measures correlate and follow a quantitatively similar pattern?

This is a good point. Inactivation of different replisome components led to reduced MCM2 phosphorylation at S53, impaired cohesin loading and cohesion defects to varying degrees. However, the MCM2 S53 phosphorylation levels did not appear to quantitatively correlate with the degrees of cohesion defects. It is possible that DDK sites other than S53 are functionally important. Alternatively, RPA and other replisome components directly contribute to NIPBL-cohesin binding. We have discussed this point more thoroughly in the revised manuscript.

16) Recently, it has been shown that MCM complex and cohesin co-IPed and shared a number of sites in both unsynchronized and synchronized cells. Furthermore, they are enriched at replication origins (Cucco et al., 2017 Nucleic Acids Res. Nov 20. doi: 10.1093/nar/gkx1172). These data should be discussed.

We have cited and discussed this recent study in the revised manuscript.

17) Figure 2. Inputs of CDC7 and MCM2 are lacking.

Because we had to blot many cohesin and replication components in many experiments, not all inputs are blotted in each experiment to save reagents. We hope that the reviewer will understand. Importantly, depletion of NIPBL, MAU2, or cohesin does not affect the total levels of CDC7 or MCM2.

18) Figures 3B and 3F are confusing. They show the same data (with the exception of XL413). Can the authors arrange them into a single figure?

These experiments were performed at different times. Not all blots are identical between these experiments. It will be difficult to combine them together into the same figure without splicing the gels. We wish to keep them separate and hope that the reviewer will understand. Certain repetition of data presentation might be beneficial to the readers.

19) Figure 3F. Inputs of ESCO, SA2, SCC4, pMCM2 and MCM5 are lacking.

We have added the input lanes to this figure.

20) Since somatic cells were used, SMC1 should be SMC1A.

As suggested, we have changed SMC1 to SMC1A in the figures.

21) Gene nomenclature should be referred as human.

We have used the proper human gene and protein nomenclature throughout the text and figures, with the exception of the Discussion and the model figures. We wish to keep the SCC2/4 nomenclature in those places, as it more widely used in the field to refer to the cohesin loader in different organisms.